# Laser-mediated rupture of chlamydial inclusions triggers pathogen egress and host cell necrosis

Markus C. Kerr[1], Guillermo A. Gomez[1], Charles Ferguson[1], Maria C. Tanzer[2,3], James M. Murphy[2,3], Alpha S. Yap[1], Robert G. Parton[1], Wilhelmina M. Huston[4] & Rohan D. Teasdale[1]

Remarkably little is known about how intracellular pathogens exit the host cell in order to infect new hosts. Pathogenic chlamydiae egress by first rupturing their replicative niche (the inclusion) before rapidly lysing the host cell. Here we apply a laser ablation strategy to specifically disrupt the chlamydial inclusion, thereby uncoupling inclusion rupture from the subsequent cell lysis and allowing us to dissect the molecular events involved in each step. Pharmacological inhibition of host cell calpains inhibits inclusion rupture, but not subsequent cell lysis. Further, we demonstrate that inclusion rupture triggers a rapid necrotic cell death pathway independent of BAK, BAX, RIP1 and caspases. Both processes work sequentially to efficiently liberate the pathogen from the host cytoplasm, promoting secondary infection. These results reconcile the pathogen's known capacity to promote host cell survival and induce cell death.

[1] Institute for Molecular Bioscience, The University of Queensland, St Lucia, Queensland 4072, Australia. [2] The Walter and Eliza Hall Institute of Medical Research, Parkville, Victoria 3052, Australia. [3] Department of Medical Biology, University of Melbourne, Parkville, Victoria 3052, Australia. [4] School of Life Sciences, University of Technology Sydney, Ultimo, New South Wales 2007, Australia. Correspondence and requests for materials should be addressed to R.D.T. (email: R.Teasdale@uq.edu.au).

A key stage in the life cycle of all intracellular pathogens is exit from the host cell. This process, termed egress, is vital to the transmission and dissemination of the organism to new hosts. Although we have gained a deep appreciation of the means by which intracellular pathogens invade and replicate within their host cells, the mechanisms by which they exit are relatively understudied[1]. In the case of pathogens that replicate within an intracellular vacuolar niche such as *Mycobacterium tuberculosis*, *Salmonella typhi* and *typhimurium*, *Legionella pneumophila* and chlamydiae, the pathogen must escape both the limiting membrane of their replicative niche and the plasma membrane in order to infect new host cells.

*Chlamydia trachomatis* is the most prevalent sexually transmitted bacterial infection among humans and is the leading cause of infectious blindness worldwide. As an obligate intracellular pathogen, *Chlamydia* maintains exquisite control over an assortment of host cellular processes during its dimorphic growth cycle. Most prominent among these is the formation of an intracellular replicative niche from the host cell's membrane-trafficking pathways[2] and the profound pro-survival influence the pathogen promotes during the replicative phase of infection[3,4].

*Chlamydia* invades host cells as a non-replicative elementary body (EB) through the action of a Type 3 Secretion System that serves to deliver bacterial effector molecules to modulate the host's membrane-trafficking and cytoskeletal elements. Once intracellular, *Chlamydia* alters the encompassing vacuole to create its replicative niche, called an inclusion, where it transitions into its metabolically active replicative reticulate body (RB) form. During the later stages of the pathogen's life cycle, *Chlamydia* asynchronously transforms back into its EB form before it egresses from the cell by one of three independent mechanisms: exocytosis[5], extrusion of the intact chlamydial inclusion from the host cell or rupture of the inclusion immediately prior to cell lysis[6]. The extrusion mechanism is an actin-dependent process[6] recently reported to be coordinated by the actions of myosin phosphatase, myosin light chain 2, myosin light chain kinase, and myosin IIA and IIB[7] and septins[8].

Although extrusion is a conserved mechanism[9] speculated to contribute to evasion of the host immune response and long-distance dissemination[10], release of the EBs to infect new host cells ultimately necessitates lysis of both the inclusion and the limiting membrane of the cell and/or extrusion. Hybiske and Stephens[6] used the pan-cysteine protease inhibitor E-64 to demonstrate the requirement for cysteine protease activity during rupture of the inclusion and also identified that intracellular calcium was required for the subsequent lysis of the limiting membrane and release of the *Chlamydia* into the extracellular milieu. The asynchronous nature of chlamydial egress has, however, impeded further dissection of the process and remarkably little is known about the molecular events involved, particularly the identity of the cysteine proteases involved in inclusion rupture.

Also, the precise nature of the consequent rupture-induced cytotoxicity is unclear, with evidence to suggest apoptotic, pyroptotic and necrotic mechanisms. Gibellini *et al.*[11] first utilised population-based analyses to report induction of apoptosis in tissue culture cells following long-term infection with either *Chlamydia psittaci* or *C. trachomatis*. Ying *et al.*[12] demonstrated that infected cells displayed apoptotic features such as nuclear condensation and fragmentation, and positive terminal deoxynucleotidyl transferase dUTP neck end labelling (TUNEL), and attributed these features to BAK, a key regulator of apoptosis. Vats *et al.*[13] provided evidence to suggest that *C. trachomatis* induces apoptotic cell death via caspase-8, which cleaves Bcl-2-interacting protein (BID) to generate truncated tBID, in turn activating the mitochondrial apoptotic pathway through

the action of BAK and BAX[14]. Jungas *et al.*[15] reported that *Chlamydia*-infected cells are profoundly resistant to intrinsic induction of apoptosis, yet share features in common with both apoptosis and necrosis and that this is both cell type- and stage of infection-dependent. In contrast, Jorgensen *et al.*[14] reported that inclusion collapse can induce caspase 1-dependent pyroptosis. Most tellingly, Schöier *et al.*[16] reported that within a heterogeneous population of cells, non-apoptotic and apoptotic death occurs within the infected and uninfected cells, respectively, demonstrating the need for higher resolution analyses of the process independent of the influence of neighbouring cells. Although Hybiske and Stephens did not observe apoptotic phenomena during chlamydial egress in their study, the 5 min interval they utilised, for a lytic process with a typical duration of 16 min (ref. 6) (~3 frames), is probably insufficient to draw robust conclusions as to the nature of the cell death pathway involved.

Keeping this in mind, we set out to define the cellular and molecular events involved in chlamydial egress at the individual cell level. Here we use a laser ablation strategy that, when combined with time-lapse videomicroscopy, allows us to selectively disrupt the inclusion membrane in live cells and monitor the cellular events thereafter. Using this approach we initially demonstrate the barrier function of the inclusion membrane, protecting the encompassed bacteria. Second, we demonstrate that the subsequent lysis of the cell is mediated by a coordinated necrotic pathway that is BAK-, BAX-, RIP1- and caspase-independent. Third, we provide evidence that this pathway is largely mediated by the host rather than the invading pathogen, and accordingly we show that inhibition of host cell calpains affects chlamydial inclusion rupture but not subsequent cell lysis.

## Results

**Inclusion rupture is directly coupled to chlamydial egress.** We first recapitulated Hybiske and Stephens[6] observation that the inclusion ruptures immediately prior to bacterial egress. A HeLa reporter cell line stably expressing mCherry-tagged Rab25 (to monitor the integrity of chlamydial inclusions throughout the infection)[17], CFP-Histone 2-B (to highlight the nuclei) and soluble green fluorescent protein (GFP) (to mark the cytoplasm) was infected with *C. trachomatis* strain LGVII (CTL2) at an multiplicity of infection (MOI) ~0.5 and examined by time-lapse videomicroscopy. From ~36 h post infection (h p.i.) the inclusions of infected cells began to rupture in an asynchronous manner, manifest by the loss of inclusion integrity and influx of cytoplasmic GFP into the inclusion lumen (asterisk) leading to an overall dimming of the GFP fluorescence. Cell plasma membrane integrity was lost from 15–30 min post-inclusion rupture (Fig. 1a, Supplementary Movie 1). Notably, the nuclei of cells generally maintained their overall structure following inclusion rupture, condensing moderately prior to cell lysis (arrows in Fig. 1a). Although the kinetics of the lytic process post-rupture were extremely consistent, inclusion rupture was observed in a stochastic manner anywhere from 36 h p.i. (Supplementary Movie 2). Although it is recognised that cysteine protease activity is required for inclusion rupture and intracellular calcium signalling is necessary for subsequent cell lysis[6], the manifest asynchronous nature of inclusion rupture has proven refractory to more detailed investigation of the molecular events involved in chlamydial egress.

Optical dissection methods provide the means to locally microirradiate regions of cells at submicron resolutions[18]. Unlike long-pulse ultraviolet and visible lasers, femtosecond lasers that operate in the near infrared region of the spectrum produce efficient two-photon ionisation with no out-of-focus

absorption[19]. Owing to nonlinear effects around the focal volume, there is little transfer of heat or mechanical energy to surrounding structures meaning that subcellular organelles may be targeted for photodisruption without influencing underlying or overlying structures. Watanabe et al.[20] demonstrated the efficacy of this approach to selectively ablate individual mitochondria without influencing cellular viability. Accordingly, we have extended this approach by developing a multiphoton ablation system[21] to specifically rupture the chlamydial inclusion (reticle) without damaging the plasma membrane of the cell, thereby allowing us

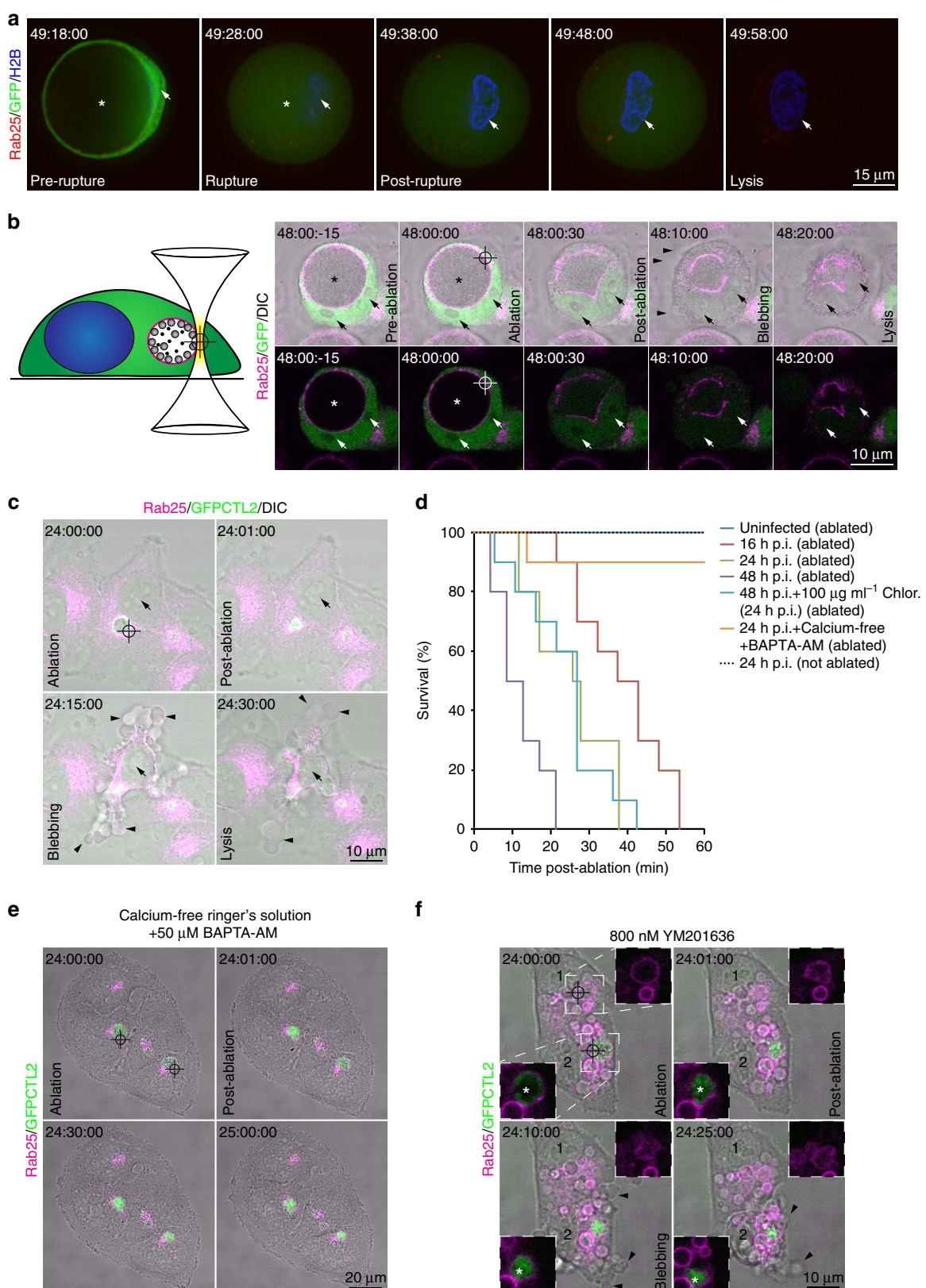

to reliably interrogate the process with high resolution and uncouple inclusion rupture from subsequent cell lysis (Fig. 1b).

HeLa cells stably expressing mCherry-Rab25 were seeded onto imaging plates, transfected with soluble GFP and infected with CTL2 for 48 h. The cells were imaged live on a line scanning confocal microscope for three frames before a $0.6\,\mu m^3$ region of the inclusion was ablated (reticle) and the samples imaged further (Fig. 1b, Supplementary Movie 3). Consistent with native egress, immediately following ablation, the inclusion is observed to fill with soluble GFP before it collapsed in a manner similar to that observed previously using mechanical disruption[22]. As expected[20], maintenance of the plasma membrane's integrity during the ablation was affirmed as the soluble GFP was observed to fill the inclusion (leading to an overall dimming) but remained within the host cell boundary. From 15–30 min following inclusion rupture, ablated cells blebbed extensively, before contracting and releasing the contents of their cytoplasm into the extracellular milieu. As in native egress, the overall structure of the nucleus is maintained with some degree of condensation (Fig. 1b, arrows).

Ablation of inclusions in HeLa cells infected for 24 h with GFP-expressing CTL2 also triggered cell death and lysis (Fig. 1c, Supplementary Movie 4). Although the initiation of cell death was modestly delayed, once initiated, the cells appeared to die with similar kinetics to those ablated at 48 h p.i. This suggested that the cell death pathway involved does not depend upon stage of infection. To examine this, the consequences of inclusion ablation for cell survival were quantified for 10 cells in the presence of cellular and bacterial inhibitors and at different stages of infection (Fig. 1d). Ablation at 16, 24 and 48 h p.i. all led to induction of cell death and lysis within 60 min with the increasing bacterial load of the later time points yielding a modest acceleration in the induction of cell death. As in native egress, the cell lysis that followed inclusion ablation was sensitive to intracellular calcium signalling[6]. Ablation of inclusions in 24 h p.i. cells pre-cultured for 1 h in calcium-free Ringer's solution supplemented with $50\,\mu M$ 1,2-bis(2-aminophenoxy)ethane-$N,N,N',N'$-tetraacetate-acetoxymethyl resulted in sustained viability of cells (Fig. 1d,e, Supplementary Movie 5). Subsequent de novo bacterial protein synthesis was not required to trigger cell death as treatment of cells with $100\,\mu g\,ml^{-1}$ chloramphenicol from 24 h p.i. did not impact upon the ablation-triggered cell death response (Fig. 1d). Importantly, the specificity of the induced cell death response to rupture of the inclusion is established by the observation that ablation of mCherry-Rab25-labelled endosomal membranes swollen in the presence of the endosomal trafficking inhibitor YM201636 (ref. 23), in both infected (Fig. 1f, cell 2) and uninfected cells (Fig. 1f, cell 1), does not trigger the cell death response (Fig. 1f, Supplementary Movie 6). Altogether, these findings suggest that it is the host that drives the lytic process, likely in response to cytoplasmic exposure to inclusion content, rather than the pathogen itself.

It was recently suggested that the secreted chlamydial protease-like activity factor (CPAF) accumulates within the chlamydial inclusion and is released upon rupture so that it may target vimentin intermediary filaments and components of the nuclear envelope for degradation[24]. Live imaging of $2\times$ GFP-tagged vimentin following inclusion ablation recapitulated this observation with a rapid dissolution of filamentous structures and a rapid translocation of $2\times$ GFP-tagged vimentin into the nucleus. A total of $2\times$ GFP-vimentin dissolution was blocked in the presence of a CPAF inhibitor[25] following inclusion rupture (Supplementary Fig. 1, Supplementary Movies 7 and 8). Consistent with observations made using a CPAF-null chlamydial strain[24] inhibition of CPAF activity, however, did not block cell lysis following inclusion rupture.

Given that all previously described molecular phenotypes attributed to chlamydial inclusion rupture and egress are maintained through our approach, laser ablation of chlamydial inclusions therefore represents a directed means to trigger chlamydial egress in a controlled fashion uncoupled from the contribution of the pathogen itself. Furthermore, inclusion ablation mid-way through the intracellular development of the pathogen allows one to examine the subsequent molecular events involved with unparalleled spatial and temporal resolution devoid of the confounding metabolic and morphological impacts the pathogen and the inclusion place upon the host cell during the later stages of infection.

**Inclusion ablation compromises its barrier function.** Initially correlative light and electron microscopy was employed to compare the intraluminal environment of intact and ablated inclusions between and within infected cells thereby providing internal controls for the protocol. Once again, time-lapse videomicroscopy revealed the rapid condensation and immobilisation of inclusions ablated 24 h p.i. when compared with intact inclusions in the same region of interest (Fig. 2a, Supplementary Movie 9). These cells were fixed 10 min post ablation and processed for transmission electron microscopy. Electron micrographs of these particular inclusions demonstrated that, although intact inclusions present characteristic spacious arrangement of chlamydial RBs in an electron-lucent lumen (Fig. 2b, cell 1), the RBs of ablated inclusions are compacted against one another in a tight arrangement with cytosol distributed between them (Fig. 2b, cell 2). Intriguingly, there was also evidence of very rapid RB swelling in the ablated inclusions (Fig. 2b, cell 2, arrows). Given that the lumen of the chlamydial

**Figure 1 | Laser-mediated inclusion rupture triggers chlamydial egress.** (**a**) Time-lapse videomicroscopy of GFP-expressing mCherry-Rab25 and CFP-H2B stable HeLa cells imaged from 36 h p.i. with CTL2 (MOI ∼ 0.5). Presented is a single egress event captured from 49 h p.i. Arrows highlight the nucleus and asterisks the inclusion. Interval of capture was 5 min. (**b**) Diagram of inclusion ablation system and time-lapse videomicroscopy of the system applied on GFP-expressing and mCherry-Rab25 stable HeLa cells 48 h p.i. with CTL2 (MOI ∼ 0.5). Arrows highlight the nuclei, asterisks the inclusion and arrow heads blebbing of the plasma membrane. Interval of capture is as indicated. (**c**) Time-lapse videomicroscopy of mCherry-Rab25 stable HeLa cells ablated 24 h p.i. with GFP-CTL2 (MOI ∼ 0.5). Arrows highlight the nuclei and arrow heads blebbing of the plasma membrane. Interval of capture is as indicated. (**d**) Quantification of cell survival of ablated and unablated cells as described in the methods under the conditions as indicated. Ten independent movies with at least one ablated cell each for each condition were examined. (**e**) Time-lapse videomicroscopy of mCherry-Rab25 stable HeLa cells ablated either 24 h p.i. with GFP-CTL2 (MOI ∼ 0.5) in calcium-free Ringer's solution and $50\,\mu M$ BAPTA-AM. Interval of capture is as indicated. Movie is representative of at least 10 independent movies, and endosomes of dimensions similar to ablated inclusions were targeted for comparison. (**f**) Time-lapse videomicroscopy of two mCherry-Rab25 stable HeLa cells ablated 24 h p.i. with GFP-CTL2 in the presence of 800 nM YM201636 to swell endosomal compartments. Numbers highlight nuclei of individual cells. Insets highlight a ruptured Rab25-positive chlamydial inclusion denoted by the prominent GFP-expressing bacterial fluorescence within (cell 2) and a ruptured Rab25-positive endosome (cell 1). Asterisks indicate the inclusion and arrow heads highlight blebbing of the plasma membrane. Movie is representative of at least 10 independent movies, and endosomes of dimensions similar to ablated inclusions were targeted for comparison.

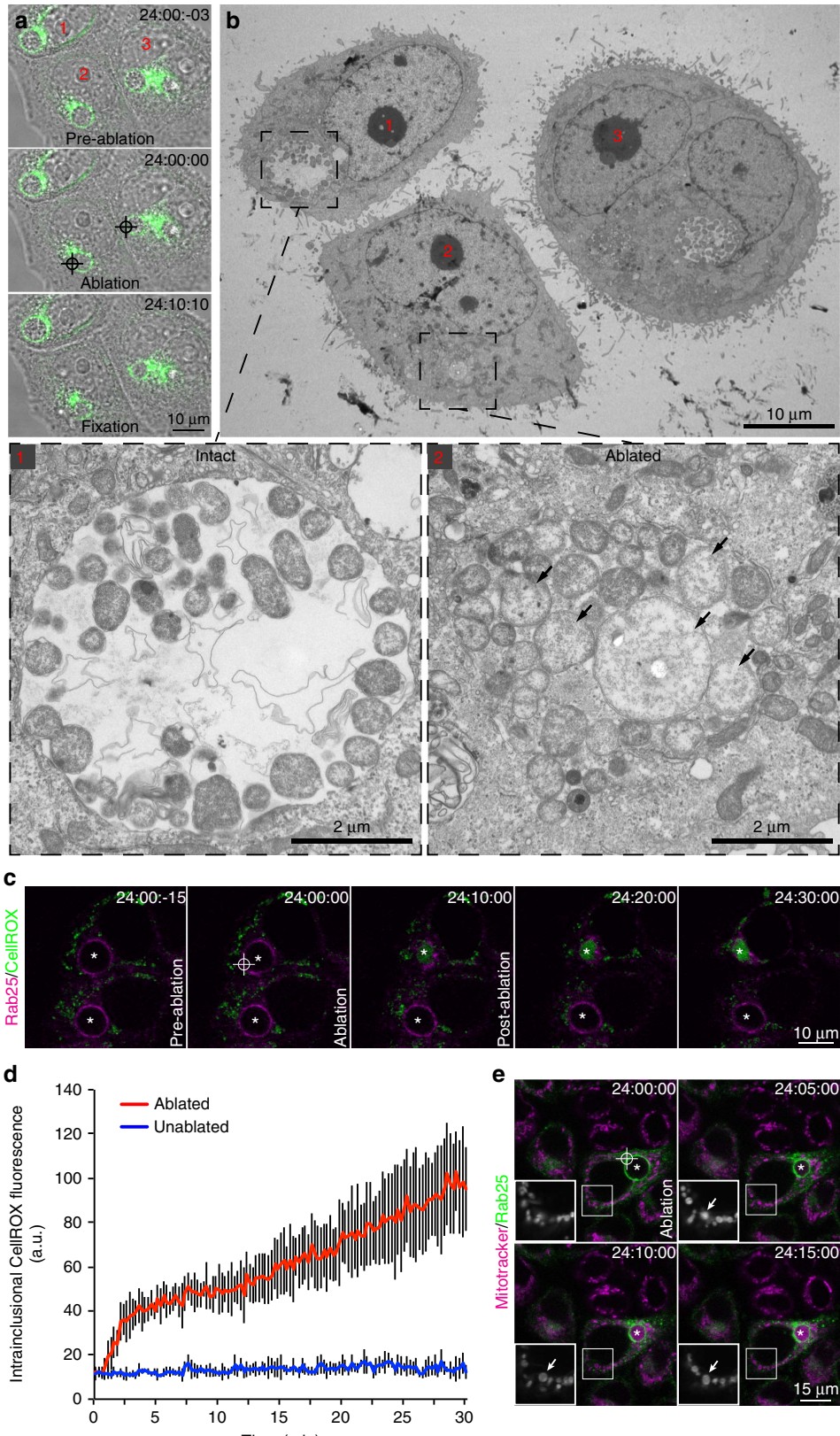

**Figure 2 | Correlative light and electron microscopy reveals the barrier function of the chlamydial inclusion.** (**a**) Time-lapse videomicroscopy of GFP-Rab25 stable HeLa cells ablated 24 h p.i. with CTL2 (MOI ∼ 0.5). (**b**) Transmission electron micrographs of these same cells. Arrows highlight swollen chlamydial RBs. (**c**) Time-lapse videomicroscopy of mCherry-Rab25 stable HeLa cells ablated 24 h p.i. with CTL2 (MOI ∼ 0.5) in the presence of CellROX Green. Arrow heads highlight intracellular mitochondrial CellROX and asterisks the inclusions. (**d**) Quantification of intra-inclusional CellROX Green fluorescence. N = 5 biological replicates. Error bars present the s.d. from the mean. (**e**) Time-lapse videomicroscopy of GFP-Rab25 expressing stable HeLa cells ablated 24 h p.i. with CTL2 (MOI ∼ 0.5) in the presence of Mitotracker Red CMXRos. Arrows highlight swelling mitochondria and asterisks the inclusion.

inclusion shares most biophysical properties in common with the cytoplasm and there is a free exchange of cytoplasmic ions[26], we considered what might contribute to this shift in bacterial morphology. Grieshaber et al.[26] proposed that the swollen shape of the chlamydial inclusion is maintained by osmotic pressure raising the possibility that inclusion rupture leads to osmotic shock. Another possibility is that of exposure to typically excluded reaction oxygen species (ROSs) now freely accessing the bacteria from the cytoplasm. Indeed, infection with C. trachomatis is recognised to induce the production of ROSs early in infection with subsequent inaction of the host's NADPH oxidase, suggesting that the pathogen actively suppresses ROS production to promote infection[27]. Paradoxically, inhibiting ROS production during chlamydial infections also suppresses the growth of the bacteria suggesting that the pathogen requires host-derived ROS for optimal growth[28].

To examine this more directly, CTL2-infected mCherry-Rab25 cells were imaged live and the inclusion membranes ablated 24 h p.i. in the presence of CellROX Green, a fluorogenic probe that exhibits bright fluorescence upon oxidation by ROSs and binding to DNA. It was immediately apparent that the ROS probe was not evident within the chlamydial inclusion (Fig. 2c,d, Supplementary Movie 10). Strikingly, ablation of the inclusion membrane yielded an accumulation of CellROX Green fluorescence within the lumen of the ablated inclusion indicating oxidation of the probe and binding to DNA, presumably of chlamydial origin. This was confirmed using the antioxidant, N-acetylcysteine[29] to quench ROS-mediated oxidation in ablated cells (data not shown). ROSs are generated as by-products of cellular metabolism, primarily in the mitochondria[30] so we used our ablation system to monitor the mitochondria of cells immediately following inclusion rupture (Fig. 2e, Supplementary Movie 11). We observed swelling and eventual dimming of Mitotracker Red CMXRos-stained mitochondria (arrows) following inclusion ablation suggesting a transition in mitochondrial permeability. In addition, we observed accumulation of the Mitotracker fluorescence within the inclusion following inclusion rupture indicating the now ready access of the thiol-reactive dye to the pathogen once the inclusion membrane is compromised. Although we cannot formally exclude the possibility that the membrane permeant probes used here are simply excluded from the inclusion lumen, its limiting membrane is derived from host cellular membranes that both probes are readily permeant to and is freely permeable to cytoplasmic ions[26]. Regardless, taken together, our results provide evidence that the chlamydial inclusion serves as a protective barrier to the hostile host cytoplasm.

**Inclusion rupture-induced cell death is not apoptotic**. The calcium-dependent, highly coordinated nature of the rupture-induced cell death is suggestive of a programmed cell death (PCD) pathway[31,32]. There are, however, discordant views on the contribution of PCD in chlamydial infection biology. Although Vats et al.'[13] findings suggest that caspase-dependent PCD, or classical apoptosis, is prominent among Chlamydia-infected primary cervical epithelial cells, Schoier et al.[16] reported that apoptosis does not appear to be the primary mode of death for infected cells within in vitro culture systems, rather it is prominent amongst the neighbouring uninfected cells. We demonstrate that the nuclear ultrastructure of cells in which inclusions have ruptured appears to be, for the most part, indistinguishable from that of cells with intact inclusions (compare nuclei of cell 1 with cells 2 and 3 of Fig. 2b), lacking the characteristic condensation and peripheralisation of chromatin (pyknosis) observed in classically apoptotic cells. They do, however, present blebbing of the plasma membrane, cleavage of vimentin (Supplementary Fig. 1) and swelling of

mitochondria (Fig. 2e), which are all prominent features of apoptosis[33]. Indeed there is significant debate in the literature as to the contribution of apoptosis to chlamydial infection-induced cell death[11–13,15,16,25,34]. This prompted us to investigate the mechanism by which the cells were dying more thoroughly.

The caspases are a family of cysteine proteases integral to apoptosis, and cysteine protease activity has been demonstrated to be required for inclusion rupture and consequently chlamydial egress[6]. Our laser-mediated approach also allows us to uncouple the events both prior to and following inclusion rupture with high fidelity. Caspases 3 and 7 are frequently activated death proteases catalysing the cleavage of many cellular proteins. To monitor caspase-3/7 activity in cells in which the inclusion had been ruptured we initially employed the CellEvent Caspase-3/7 Green detection reagent in combination with time-lapse videomicroscopy (Fig. 3a, Supplementary Movie 12). To ensure the probe was functional, the nuclei of individual cells were also ablated to induced classical apoptosis (Fig. 3a, Supplementary Movies 12 and 13). Strikingly, although the cells in which the nuclei had been targeted rapidly became fluorescent in the 488-channel indicating cleavage of the Caspase-3/7 Green probe, those cells in which the inclusions had been ruptured and subsequently died remained resolutely devoid of any detectable fluorescence indicating that caspases 3 and 7 were not activated. Furthermore, although treatment of cells with the pan-caspase inhibitor Z-VAD-fmk (50 μM) was sufficient to inhibit cell death and CellEvent Caspase-3/7 Green activation in nuclear ablated cells (Supplementary Movie 14), it did not inhibit ablation-triggered cell death (Fig. 3b) or native egress of the pathogen (Fig. 3c). Similarly, treatment with the caspase 1 selective inhibitor, VX765 (50 μM), did not inhibit inclusion rupture-triggered death (Supplementary Fig. 2a) or native egress of the pathogen (Supplementary Fig. 2b).

Activation of caspase-3 requires proteolytic processing of its inactive zymogen into activated p17 and p12 fragments[35]. As our ablation approach is refractory to population-based analyses like western blotting to assess the abundance of these fragments and it has been established that chlamydial infection induces apoptosis in neighbouring uninfected cells, we applied correlative live cell and immunofluorescence microscopy with a cleaved caspase-3 specific antibody to determine whether or not inclusion rupture leads to activation of endogenous caspase-3 (Supplementary Fig. 3a,b, Supplementary Movie 15). Consistent with the observations made with the Caspase-3/7 Green probe, no evidence of endogenous caspase-3 cleavage[35] was observed nor was there any evidence of cleavage of caspase 8 (refs 36,37) (Supplementary Fig. 3c,d, Supplementary Movie 16). Similarly, TUNEL labelling, to monitor DNA fragmentation, indicated that classic markers of apoptosis were absent in cells with ruptured inclusions (Supplementary Fig. 4a,b and Supplementary Movie 17). Notably, laser ablation of nuclei resulted in a localised cauterisation wound within the DNA of cells that was prominently TUNEL-labelled (Supplementary Fig. 4b). This labelling extended in a more diffuse manner throughout the entire nuclei of these cells consistent with them having initiated apoptosis and DNA fragmentation. Although there was no evidence of TUNEL-labelling within the nuclei of cells in which inclusions had been ablated (or neighbouring cells not targeted), prominent TUNEL-labelling was observed within Chlamydia (arrow heads) in which inclusions had been disrupted consistent with the hypothesis that inclusion disruption exposes the bacteria within to the damaging cytoplasm of host cells. The discord with previous observations that report positive TUNEL-labelling in the late stages of chlamydial infection[12] most likely reflects confounding intercellular signalling consequences consistent with observations made by Schoier et al.[16]

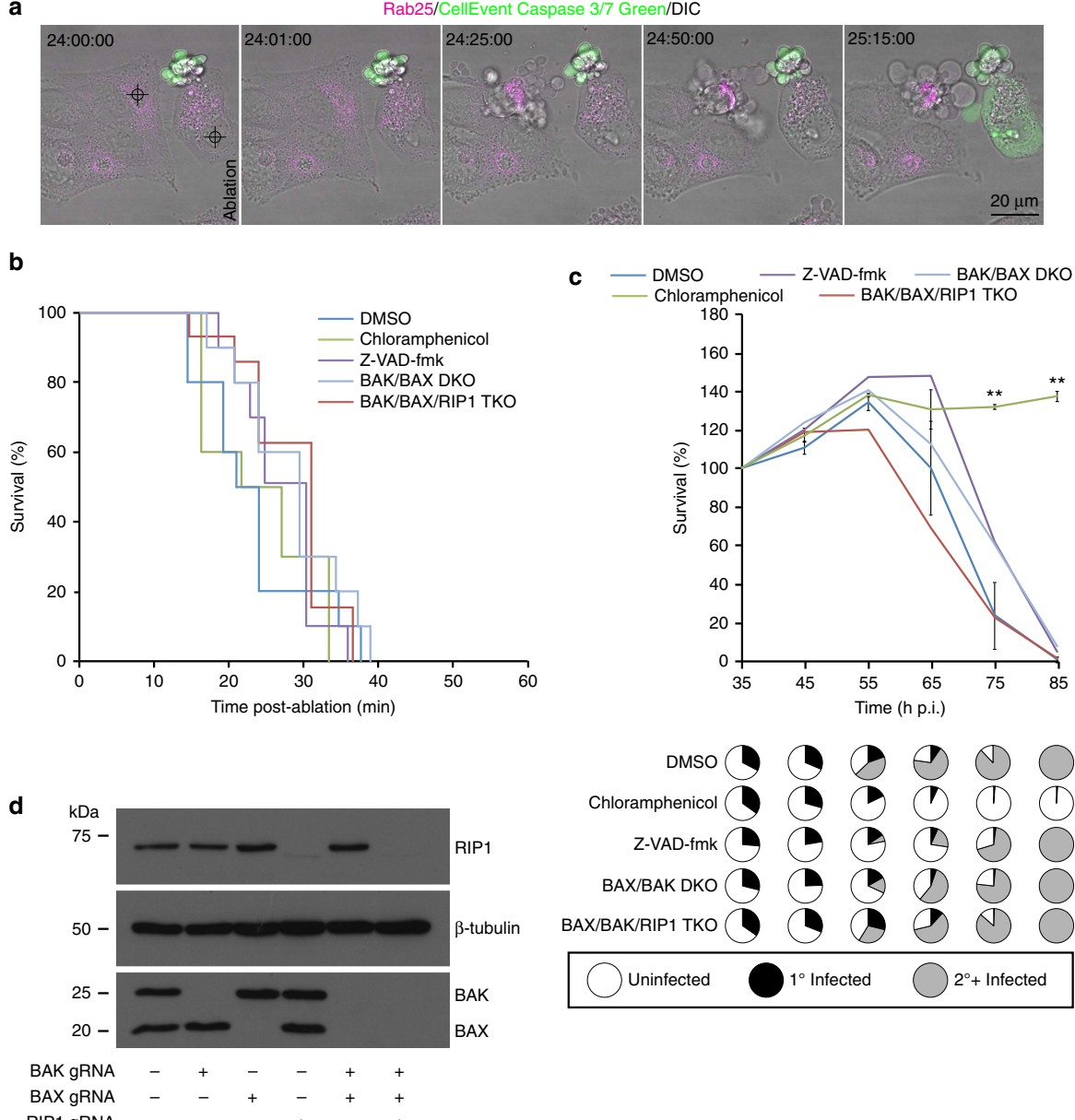

**Figure 3 | Chlamydia-induced cell death is not apoptotic or necroptotic.** (**a**) Time-lapse videomicroscopy of mCherry-Rab25 stable HeLa cells ablated 24 h p.i. with CTL2 (MOI ∼ 0.5) in the presence of CellEvents Caspase-3/7 Green Detection Reagent. (**b**) Quantification of inclusion rupture induced cell death under the indicated conditions. $N = 10$ biological replicates for each condition. (**c**) Quantification of native egress-induced cell death under the indicated conditions. Presented in pie-charts are the proportions of surviving primary infected, secondary infected and uninfected cells at the indicated time-point as monitored by GFP-CTL2 fluorescence. $N = 3$ biological replicates with > 200 cells counted per replicate. Error bars present the s.d. from the mean. For clarity, only error bars and $P$ values for DMSO and chloramphenicol are presented. $**P ≤ 0.01$; (Unpaired Student's $t$-test). (**d**) Western blotting of HeLa cells that had been genome-edited for the indicated targets full blots are shown in Supplementary Fig. 5.

The Bcl-2 family of proteins constitute critical control points in the intrinsic and extrinsic apoptotic pathways[38]. Ordinarily, pro-apoptotic BAX resides in the cytosol or is loosely associated with membranes but, in response to death signals, is inserted into the mitochondrial outer membrane as a homo-oligomerised multimer, resulting in mitochondrial dysfunction. Similarly, in healthy cells, an interaction between the voltage-dependent anion channel protein 2 with inactive pro-apoptotic BAK keeps this lethal molecule in check at the mitochondrion. In response to stress signals, this interaction is displaced allowing BAK to homo-oligomerise and cause mitchondrial dysfunction. In both cases mitochondrial outer membrane permeabilisation results in

leakage of mitochondrial content, which in turn activates initiator and downstream effector caspases. To definitively rule out apoptosis and to examine whether the observed mitochondrial swelling was a consequence of BAK and/or BAX activity, we applied CRISPR technology to block formation of the mitochondrial outer membrane permeabilisation by selectively disrupting *BAK* and/or *BAX* in HeLa cells (Fig. 3d, Supplementary Fig. 5). Cells knocked out for BAK and BAX are known to be resistant to intrinsic and extrinsic apoptotic stimuli as well as selected necrotic stimuli[39]. Consistent with these observations our BAK/BAX double-knockout cells as well as those treated with Z-VAD-fmk showed complete insensitivity to

 7

extrinsic and intrinsic apoptotic stimuli (Supplementary Fig. 6). Interestingly, however, ablation-triggered and native egress were not inhibited in these cells (Fig. 3c), nor was there any evidence of mitochondrial translocation of GFP-BAX[40] following inclusion ablation (Supplementary Movie 18). Taken together, the events subsequent to chlamydial inclusion rupture and ultimately egress of the pathogen are not BAK/BAX- or caspase-dependent and therefore do not constitute classical apoptosis.

**Chlamydial egress is not mediated by a necroptotic pathway**. Given the apparently caspase-independent nature of chlamydial inclusion rupture-induced PCD, we next investigated the like-lihood that the mechanism may be necrotic. The highly respon-sive and triggerable nature of the death as well as its dependence upon intracellular calcium suggested that the pathway could be programmed necrosis or 'necroptosis'[41]. Necroptosis occurs when cells, in which apoptosis signalling has been blocked, are exposed to death-inducing stimuli and is mediated through the action of kinase Receptor Interacting Proteins (RIPs) 1 and 3. RIP1 and RIP3 form supramolecular complexes called the necrosomes, which in turn lead to the activation of the mixed lineage kinase domain-like protein by RIP3-mediated phosphorylation causing it to homo-oligomerise[42,43]. Oligomerised mixed lineage kinase domain-like protein translocates to the plasma membrane of the cell where it compromises its ability to preserve intracellular ionic homoeostasis. Necroptosis is inhibited by necrostatin-1, a small molecule inhibitor of RIP1 kinase[44]. To determine whether the cell death pathway in question is necroptosis, ablation-triggered and native egress were examined in the presence of this agent and the rate at which cells died monitored. Interestingly, treatment with 100 µM necrostatin-1 did not delay cell death in either circumstance (Supplementary Fig. 2). To definitively rule out necroptosis, CRISPR-mediated knockout of RIP1 in HeLa and BAK/BAX double-knockout HeLa cells was performed (Fig. 3d) and both ablation-triggered and native egress examined (Fig. 3b,c). Similar to the observations made earlier, ablation-triggered and native egress progressed normally in these cells. The egress of the pathogen is therefore distinct from the established apoptotic and necroptotic PCDs.

**Calpain inhibitors affect inclusion rupture**. Our laser ablation approach allows us to distinguish between the molecular events involved in inclusion rupture from those involved in cell lysis. Calpeptin is a membrane permeable active-site directed inhibitor of calpains[45]. Intriguingly, unlike Z-VAD-fmk, treatment of *C. trachomatis*-infected cells with 100 µM calpeptin (Fig. 4a, Supplementary Movies 19 and 20) led to maintenance of chlamydial inclusion integrity, resulting in much larger inclusions when compared with control infections. In spite of calpain inhibition, eventual inclusion rupture triggered cell lysis (Fig. 4b) but did not perturb cell death following laser ablation (Fig. 4c), indicating that calpain activity may only be required for the first stage of chlamydial egress. Similar observations were made using the broader acting calpain inhibitor 100 µM l-*trans*-3-ethoxycarbonyloxirane-2-carbonyl-l-Leu-(3-methylbutyl)amide. Liberation of EBs into the extracellular milieu was measured by conducting infectious progeny assays using media sampled from cells in the presence and absence of 100 µM calpeptin at 72 h p.i. (Fig. 4d,e). A marked reduction in extracellular EBs was observed in calpain-inhibited cells when compared with those treated with the carrier solvent. Notably, cell lysates treated with calpain inhibitors presented similar numbers of infectious EBs, indicating that calpain inhibition did not adversely impact the pathogen directly (Fig. 4d,e). Therefore, calpains may be required for

chlamydial inclusion rupture but not the consequent cell death or the pathogen's growth.

## Discussion

In order to propagate an infection, the obligate intracellular pathogen, *C. trachomatis*, must lyse its host cell or enveloping extrusion in order to infect new cells. The resultant cellular damage and inflammation is likely responsible for the acute pelvic inflammatory disease, scarring and potential infertility, frequently associated with infection. Yet, it has long been recognised that *Chlamydia*-infected cells are insensitive to both extrinsic and intrinsic apoptotic stimuli[3] and this resistance is required for the pathogen to complete its unique dimorphic life cycle. The resistance to apoptosis appears to be acquired by a variety of mechanisms including degradation of a key regulator of apoptosis, p53 (refs 46,47), but notably, *Chlamydia*-infected HeLa cells exhibit a marked reduction in caspase 8 enzyme activity[48]. Activated caspase 8 propagates apoptotic signalling by either directly cleaving and activating downstream effector caspases or by cleaving the BH3-only protein, BID, which translocates to the mitochondria and induces leakage of cytochrome c. We, however, find no evidence of effector caspase activation in infected cells during native chlamydial egress or following inclusion rupture, and treatment with pan-caspase inhibitors does little to preserve cell or inclusion integrity of infected cells (Fig. 3a–c). We did, however, observe preservation of neighbouring uninfected cell integrity during chlamydial egress when caspase activity was inhibited (Fig. 3c and Supplementary Fig. 2b) consistent with *in vivo* observations that suggest a role for caspase activity in *Chlamydia*-induced infertility[49]. Indeed, although caspase 1-deficient mice display much reduced genital tract inflammatory damage following chlamydial infection, the mice experience similar courses of infection, indicating that caspase 1 does not have a direct role in the establishment and progression of the infection[50].

How then does *Chlamydia* kill host cells? In contrast to Perfettini *et al.*[51], who observed that BAX translocates to mitochondria in *C. psittaci*-infected cells and that BAX-inhibitor 1 or Bcl-2 expression perturbs *C. psittaci*-induced caspase-independent 'apoptosis', here we find that HeLa cells rendered profoundly resistant to apoptosis and/or necroptosis through the selected and iterative deletion of BAK, BAX and RIP1 as well as in the presence of pan-caspase and necroptosis inhibitors still die with the same kinetics as control cells (Fig. 3b,c). Whether this disparity with the work of Perfettini *et al.*[51] reflects a biological variance unique to the different species of *Chlamydia* used in the respective studies remains to be clarified but our data indicate that the pathway in question is not apoptotic. Recently, RIP3-independent necroptotic pathways have been described[52], including one that is induced in response to alkylating DNA-damage agents and involves the sequential activation of poly(ADP-ribose) polymerase 1, calpains, BAX and AIF[53]. Intriguingly, DNA strand breaks are associated with loss of plasma membrane integrity and organelle dilation at the later stages of the infection with *Chlamydia pneumoniae*[54]. The chlamydial inclusion rupture-triggered cell death is, however, distinct from this pathway as HeLa cells do not express RIP3 endogenously[55] and undergo death independently of BAX (Fig. 3b,c).

We find that *Chlamydia*-induced cell death is calcium-dependent, occurs regardless of the point of the infectious cycle the chlamydial inclusion ruptures, and persists in the presence of bacterial protein synthesis inhibitors. Together, these results suggest that the lytic process is largely directed by the host cell (Figs 1d and 3c). Notably, application of a small molecule

inhibitor of the CPAF had no impact on inclusion rupture or cell death and lysis, but served to maintain the integrity of host cell intermediate filaments as observed by monitoring vimentin-2×

GFP in intact live cells (Supplementary Fig. 1). There has been significant controversy surrounding the action of this particular chlamydial protein because of a post-cell lysis *in vitro* artefact[56].

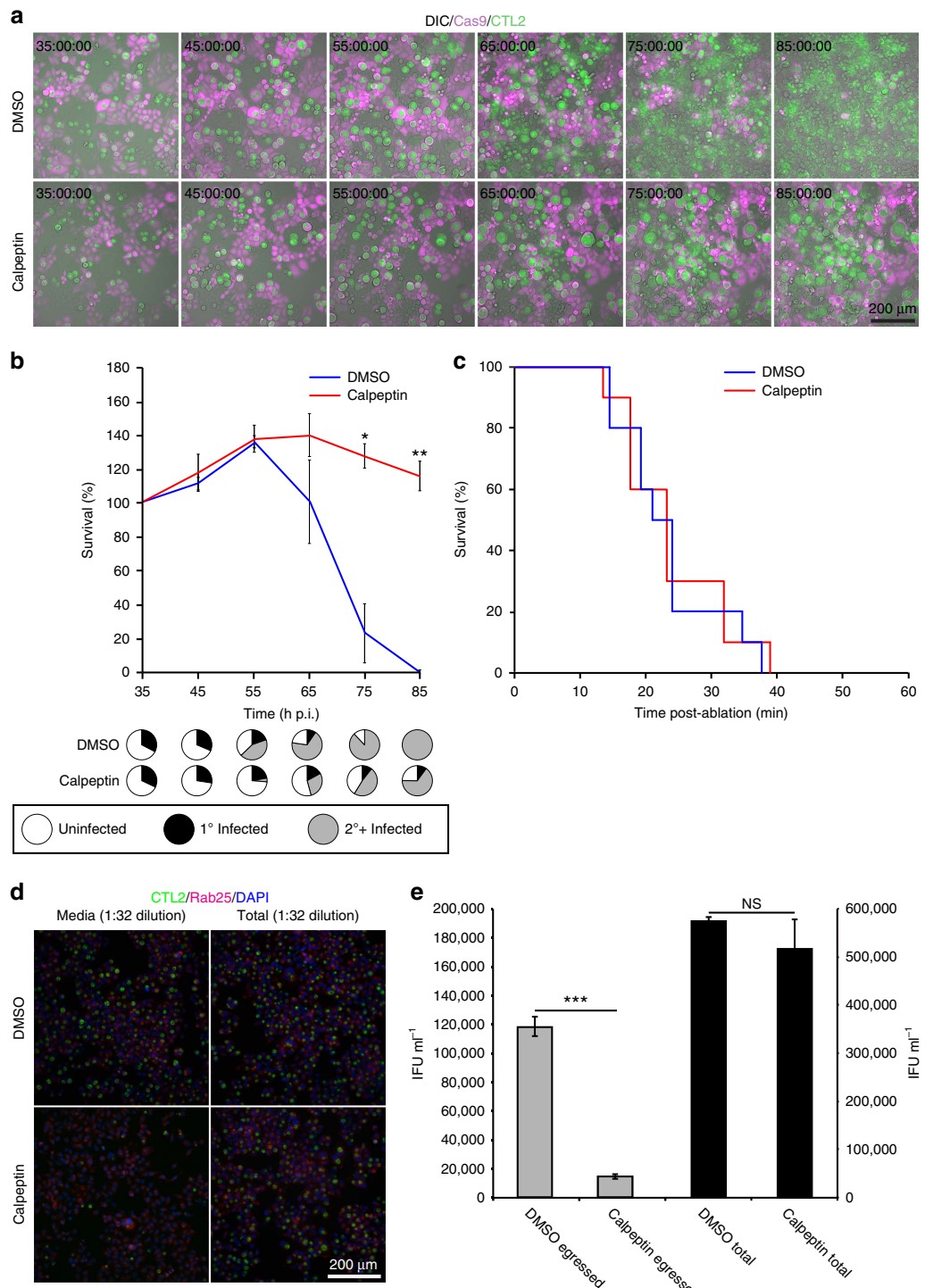

**Figure 4 | Calpain inhibitors inhibit inclusion rupture. (a)** Time-lapse videomicroscopy of mCherry-Cas9 stable HeLa cells infected from 8–72 h p.i. with GFP-CTL2 (MOI ∼ 0.5) in the presence or absence of 100 μM calpeptin. **(b)** Quantification of native egress-induced cell death under the indicated conditions. Presented in pie-charts are the proportions of primary infected, secondary infected and uninfected cells surviving cells at the indicated time point as monitored by GFP-CTL2 fluorescence. N = 3 biological replicates with > 200 cells counted per replicate. Error bars represent the s.d. the mean. *$P \leq 0.05$; **$P \leq 0.01$; ***$P \leq 0.001$; NS, Not Significant (Unpaired Student's *t*-test). **(c)** Quantification of inclusion rupture induced cell death under the indicated conditions. N = 10 biological replicates for each condition. **(d)** mCherry-Rab25 and CFP-H2B stable HeLa cells infected under the conditions as indicated from HeLa cells infected for 72 h p.i. with GFP-CTL2 (MOI ∼ 0.5). Presented images were captured 24 h p.i. **(e)** Infectious progeny-forming units in media and whole cell lysates from infected cells cultured in the presence or absence of calpeptin were quantified. N = 3, Error bars represent the s.d. from the mean. *$P \leq 0.05$; **$P \leq 0.01$; ***$P \leq 0.001$ (unpaired Student's *t*-test).

Although degradation of vimentin was clearly refuted[56], a more recent publication in which CPAF was genetically deleted from the pathogen provided evidence for a role in post-inclusion rupture disassembly of vimentin-positive intermediate filaments[24]. Our observations support this conclusion.

What precisely is released from the inclusion following rupture to trigger the events that lead to cell death remains unknown. Indeed the chlamydial inclusion is a unique environment that contains an assortment of cellular and bacterial material. Among these are likely a huge array of Pathogen-associated molecular patterns and danger signals, many of which could trigger a necrotic response like that observed in this study[57]. Necrosis is marked by an elevation in ROSs and mitochondrial hyperpolarisation but is independent of both caspase activation and RIP1 (ref. 58): all features observed or consistent with our observations following chlamydial inclusion rupture.

Finally, although inhibition of host caspases had no impact upon inclusion rupture or cell lysis, inhibition of host cell calpains lead to a marked extension in inclusion integrity without interfering with cell death, indicating that calpain-activation may be required for chlamydial inclusion lysis. Calpains are a large family of ubiquitous calcium-sensitive non-lysosomal cysteine proteases responsible for a wide variety of cellular processes[59]. Our observation is therefore consistent with and extends upon Hybiske and Stephens' original observation that application of a pan-cysteine protease inhibitor cocktail inhibited inclusion rupture[6]. Following autocatalytic hydrolytic activation, calpains translocate to intracellular membranes where they cleave a diverse suite of substrates including cytoskeletal and adhesion proteins, various membrane proteins, kinases, phosphatases, ion transporters and phospholipases[60]. Further investigation should identify the relevant substrates cleaved by the calpains during chlamydial inclusion rupture and whether they represent a viable target for therapeutic intervention during chlamydial infection.

In summary, we have developed a laser ablation method that enables us to trigger the ordinarily stochastic rupture of the chlamydial inclusion in a controlled and predictable manner. This enabled us to segregate the molecular events and contributions of both the pathogen and the host during chlamydial egress with unparalleled spatiotemporal resolution. First, we show that pharmacological inhibition of host calpains inhibit the rupture of the chlamydial inclusion but not the subsequent lytic cell death pathway, opening new avenues for investigation. Second, we demonstrate that (at least in our experimental system) the lytic cell death pathway observed in Chlamydia-infected cells occurs independent of stage of infection, BAK, BAX, RIP1 and caspase activities and provide further evidence for a role for the chlamydial effector CPAF in the disassembly of intermediate filaments following inclusion rupture. Finally, our method, which could be applied for any vacuolar pathogen, highlights the barrier function played by the inclusion membrane, protecting the encompassed bacteria from hostile cytoplasmic elements.

## Methods

**Constructs and reagents.** 1,2-bis(2-aminophenoxy)ethane-$N,N,N',N'$-tetraacetate-acetoxymethyl ester, fluorescently-labelled secondary antibodies, CellEvent Caspase-3/7 Green Detection reagent, CellROX Green and MitoTracker Red were supplied by Thermo Fischer Scientific. Calpeptin, PD150606, Necrostatin-1, carbobenzoxy-valyl-alanyl-aspartyl-[O-methyl]- fluoromethylketone (Z-VAD-fmk) and VX-765 were supplied by Merck Millipore. CPAF inhibitor peptide >95% purity was supplied by Genscript. 4',6-diamidino-2-phenylindole, paraformaldehyde and $N$-acetyl-L-cysteine were supplied by Sigma Aldrich. H2B-CFP used was as described previously[61], Addgene plasmids: 25,998. BAK (D4E4, #12,105, 1:1,000), BAX (D2E11, #5,023, 1:1,000) cleaved caspase-3 (5A1E, #9,664, 1:100), cleaved caspase 8 (18C8, #9,496, 1:100) and RIP1 (D94C12, #3,493, 1:1,000) polyclonal antibodies were supplied by Cell Signalling Technology. β-tubulin antibodies were supplied by Li-Cor (#926-42,211, 1:2,000).

**Cell culture and generation of edited lines.** HeLa cells (ATCC CCL-2) were maintained in Dulbecco's Modified Eagle Medium supplemented with 10% (v/v) fetal calf serum and 2 mM L-glutamine (Invitrogen) in a humidified air/atmosphere (5% $CO_2$) at 37 °C. Cells were confirmed to be mycoplasma-free by electron microscopy. Genome-editing was performed through the sequential generation of stably expressing mCherry-Cas9 cells followed by the delivery of target specific gRNAs[62] in the FH1tUTG lentiviral vector system. BAK gRNA: 5′-GGCCATGCTGGTAGACGTGT-3′. BAX gRNA: 5′-TCTGACGGCAACTT CAACTG-3′[63]. RIP1 gRNA: 5′-AGTGCAGAACTGGACAGCGG-3′. Clonal populations of cells were isolated by Fluorescence Assisted Cell Sorting.

**Chlamydial infections and infectious progeny assay.** GFP-expressing C. trachomatis serovar L2 (GFP-CTL2) were generated from C. trachomatis serovar L2 (CTL2, ATCC VR-902B) as described previously[64]. Cells were infected at the indicated MOI; after 2 h the media was replaced with fresh growth media, and the cells were grown to the stipulated time points. Infectious progeny assays were performed by lysing the infected cells in 0.06% NP-40 (diluted in media) and serially diluting the resulting lysates as previously described[46]. The dilutions were used to infect HeLa cell monolayers in a 96 well format for 2 h before the media was replaced with fresh growth media and the infected cells cultured for a further 22 h before fixation and imaging. In all cases, the number of chlamydial inclusions were enumerated using the cell counter plugin Fiji 1.47i (http://imagej.nih.gov). Primary infections were designated as those that were a consequence of the initial round of infection. Secondary infections were those that were evident to have formed subsequently as monitored using time-lapse videomicroscopy.

**Time-lapse videomicroscopy.** For long-term live cell imaging, monolayers were cultured in 35 mm glass-bottom dishes (MatTek) or 96 well glass-bottom micro-plates. Time-lapse videomicroscopy was carried out on individual live cells using a Nikon Ti-E inverted deconvolution microscope using a × 40, 0.9 Plan Apo DIC objective, a Hamamatsu Flash 4.0 4 Mp sCMOS monochrome camera and 37 °C incubated chamber with 5% $CO_2$. CFP fluorescence was excited with a 438/24 nm LED and captured using a 579/40 nm emission filter, GFP fluorescence was excited with a 485/20 nm LED and captured using a 525/30 nm emission filter, mCherry fluorescence was excited using a 560/25 nm LED and captured using a 607/36 nm emission filter.

**Laser ablation and confocal time-lapse videomicroscopy.** Ablation experiments were performed on an LSM 510 meta Zeiss confocal microscope at 37 °C. Images were acquired using a × 63 objective, 1.4 NA oil Plan Apochromat immersion lens at × 1.5 digital magnification, with the pinhole adjusted to three Airy units to obtain optical sections 2 μm thick. Time-lapse images were acquired before (3–5 frames) and after ablation with an interval and duration as indicated. A Ti:sapphire laser (Chameleon Ultra, Coherent Scientific) tuned to 790 nm was used to ablate subcellular regions using a constant ROI ($\sim 0.6$ μm$^2$) with one iteration at 65% transmission. The Z-dimension of our multiphoton imaging system is predicted to be 0.6 μm as calculated using a Nyquist rate and PSF calculator (https://svi.nl/NyquistCalculator), although the point spread function could be as large as 1–1.5 μm meaning that the region ablated is estimated to be $\sim 0.6$–1.9 μm$^3$. GFP and mCherry fluorescence intensities were monitored before and after the ablation using a 488 nm or 543 nm laser for excitation and a 500–550 nm or >560 nm emission filters, respectively.

**Correlative light and electron microscopy.** For correlative light and electron microscopy, cells were seeded and imaged on gridded glass-bottom dishes (MatTek). Prior to high resolution time-lapse videomicroscopy, low magnification images of the cells to be examined were captured to obtain their coordinates so that they could be identified during EM processing. Immediately post-acquisition, cells were fixed in 2.5% glutaraldehyde in PBS and processed for flat embedding in resin[65]. After curing, the resin containing the cells was broken away from the plastic dish. Cells of interest were located by reference to the grid coordinates transferred onto the resin. 60 nm sections were cut parallel to the substratum and imaged after on-grid staining in a Jeol (Tokyo, Japan) 1,011 transmission electron microscope.

**Western blotting.** Infected and uninfected cell monolayers were lysed directly with sodium dodecyl sulfate (SDS) lysis buffer (100 mM Tris/HCL, pH 6.8, 4% SDS, 20% glycerol, 0.02% bromophenol blue, 200 nM dithhiothreitol) and immediately boiled at 95 °C for 10 min. Equal amounts of protein were loaded, resolved on 10% SDS-polyacrylamide gels and transferred onto Immobilon-FL polyvinylidene difluoride membranes (Millipore, USA) according to the manufacturer's instructions. Western blotting using ECL was performed as described previously[66] using the antibodies and dilutions as described in the constructs and reagents section of the Methods.

**Apoptosis and cell survival assays.** Extrinsic or intrinsic apoptosis of HeLa cells was induced through the application of either 50 ng ml$^{-1}$ recombinant human

TNFα (ORF Genetics) and 10 μg ml$^{-1}$ cycloheximide (Sigma Aldrich) or 2 μg ml$^{-1}$ staurosporine (Sigma Aldrich), and imaged live at 24 h. Alternatively, cells were infected with GFP-CTL2 (MOI ∼ 0.5) and imaged using time-lapse videomicroscopy from 35 hp.i. Cell viability was established by monitoring the containment of soluble mCherry-Cas9 and/or GFP-CTL2 using a Nikon Ti-E inverted deconvolution microscope using a × 40, 0.9 Plan Apo DIC objective, a Hamamatsu Flash 4.0 4 Mp sCMOS monochrome camera. Cells that had lost their cytoplasmic content were scored as dead/non-viable. Statistical analyses were conducted using Excel and Prism (GraphPad). Error bars represent means ± s.d. $P$ values were determined using an unpaired Student's $t$-test; values less than 0.05 were considered statistically significant. Live cell imaging of caspase-3/7 activity was conducted using CellEvent Caspase-3/7 Green Detection Reagent as per manufacturer's instructions (Thermo Fischer Scientific).

**Statistical analysis.** Statistical analyses were conducted using Excel and Prism (GraphPad). Error bars represent means ± s.d. of 3–5 replicates (as indicated in appropriate figure legend). $P$ values were determined using an unpaired Student's $t$-test; values $< 0.05$ were considered statistically significant.

**Data availability.** The authors declare that all the relevant data supporting the findings of this study are available within the article and its Supplementary Information Files, or from the corresponding author upon request.

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

## Acknowledgements

This work was supported by funding from the National Health and Medical Research Council (NHMRC) of Australia (606788, APP1105754, APP1044041, APP569542, APP1041929, APP1037320, APP1067405, 9000220), Victorian Government Operational Infrastructure Support schemes and the Australian Research Council (DP150100364). M.C.K. was supported by an Australian Research Council Discovery Early Career Researcher Award (DE120102321) and M.C.T. was supported by a Victorian International Research Scholarship. We acknowledge the facilities and the scientific and technical assistance of the Australian Microscopy & Microanalysis Research Facility, UQ and the Australian Cancer Research Foundation (ACRF)/Institute for Molecular Bioscience (IMB) Dynamic Imaging Facility for Cancer Biology. Finally, we thank David Segal, David Huang, John Silke and Marco Herold (The Walter and Eliza Hall Institute of Medical Research, Australia) for reagents and helpful discussions in the preparation of this manuscript.

## Author contributions

M.C.K. designed and conducted all of the experiments, wrote the manuscript and constructed the figures. G.A.G. and A.S.Y. assisted in the development and optimisation of the laser ablation protocol employed throughout the manuscript and provided useful input into the manuscript. C.F. and R.G.P. conducted the correlative light and electron microscopy presented in Fig. 2. M.C.T. and J.M.M. developed, optimised and supplied the CRISPr and gRNA technology presented in Fig. 3. W.M.H. provided *C. trachomatis* strains and useful input into the manuscript. R.D.T. designed experiments and wrote the manuscript.

## Additional information

**Competing financial interests:** The authors declare no competing financial interests.

