## [Peer Review File · Nature Communications]

Reviewers' comments:

Reviewer #1 (Remarks to the Author):

Previous studies have shown that the chlamydial inclusion can be extruded from infected cells, and also that cell death of infected cells is affected by the infection. The authors propose the novel view that rupture of the chlamydial inclusion could affect exit of the bacteria from infected cells and also affect host cell death.

The inclusion ablation technique is interesting, but some points would benefit from clarification. The authors write that after inclusion ablation there is "evidence of RB swelling in the ablated inclusions suggestive of a destructive impact of inclusion rupture upon the pathogen", but RBs swell inside intact inclusions when their development is impaired (after treatment of infected cells with IFN γ or adenosine, for example). It's not obvious that the pathogen is being destroyed.

In addition, the authors need to provide more information regarding the ablation technique in this manuscript, not just refer to their previous manuscript, since it is central to this study. It is not clear why the ablation target is a volume (0.6 μm^3), nor why the technique is considered uniformly successful at disrupting the inclusion while never damaging the plasma membrane.

It's not clear why the CellRox would indicate "dramatic accumulation" of ROS in the "inclusion" after ablation. Shouldn't it be similar to the cytosol, but no longer excluded? It is implied that CellRox enters the inclusions, and ROS are excluded.

The authors refer to "the so-called Mitochondrial Outer Membrane Pore (MOMP)", but this abbreviation should be avoided if possible to avoid confusion. The chlamydiae also have a "MOMP" (major outer membrane protein) which is well known in the field.

The cellular morphology in figure 1a doesn't seem typical of a HeLa cells at 48 hpi; basically the whole cell is occupied by an inclusion in the figure. It is surprising that ablation of a 16 hpi inclusion leads uniformly to cell death within 60 min, while ablation of uninfected cells causes no cell death. It might be informative to do the experiment at an earlier time point, if they can see the inclusions to shoot at.

The statement that bacterial protein synthesis is not required for the cell lytic process is not fully supported. The authors add chloramphenicol at 24 hpi and observe no difference in host cell lysis at 48 hrs (compared to no chloramphenicol); however, they also observed no difference when they performed ablation at 24 hpi compared to 48 hpi. From the results, it's not certain whether bacterial proteins play a role in subsequent host cell lysis. The authors should tone down this conclusion or substantiate it.

Reviewer #2 (Remarks to the Author):

In this paper, Kerr et al use a laser ablation strategy to control the rupture of the *C. trachomatis* inclusion membrane, and record ensuing events by live microscopy.

The new findings of this study are:

1. inclusion rupture is inhibited by calpain inhibitors
2. cell-death that follows inclusion rupture does not require integrity of established apoptotic or necroptotic signaling cascades.

My opinion is that these data are too preliminary (point 1) or their interest is too limited (point 2) for publication in Nat. Comm..

Point 1: the calpain data are very preliminary. They only rely on the use of inhibitors, and there is

no clue on what cellular process might be implicated. Also, do the inhibitors used target the cysteine-protease activity of calpain, and how specific are they? Comparison of active concentration of the drugs, in combination or not with the cysteine inhibitor used by Hybiske et al should determine whether calpain activity is the only cysteine activity required for inclusion rupture, but event that will not answer the important question: what lies behind this calpain sensitivity?

Point 2: While there has been indeed a bit of a debate on whether infected cells undergo apoptosis or not at the end of the infectious cycle most researchers have now agreed for quite some time that apoptotic cascades are not involved (see *Curr Immunol Rev.* 2007;3(1):31-40 for instance). Finding that sudden release in the cytoplasm of hundreds of bacteria, with all the associated PAMPs, triggers rapid cell death is really not surprising, and likely involves several redundant "hyper-stress" pathways. The authors show that some canonical pro-death signaling pathways are not needed and it will likely be impossible to find a single one that is absolutely required.

The paper presents a number of shortcomings (see below). In addition, it is not focused on one message, but is more an accumulation of separate questions the authors could address with the laser technology, and fails at delivering a message. As a consequence, it lacks structure. In line with this, the discussion seems randomly organized, does not start with the important message(s), and is hard to follow. For instance the authors chose to make the calpain data the strong point of the various data they collected (see title and running title) but calpain is only introduced (very vaguely, with no reference) line 446 of the discussion, and only the last paragraph of the result section is related.

Other major remarks

1. Fig. 2c&d show that the CellROX Green probe only reacts with the inclusion lumen content after inclusion ablation. The authors interpret this as a proof that the inclusion membrane protects the bacteria from ROS. A control that the probe reached the inclusion lumen (but did not react) before inclusion ablation is needed (same for the use of Mitotracker Red).
2. The authors claim that inclusion rupture exposes bacteria to ROS, and that this damages the bacteria. The data do not support these conclusions. They only show RB swelling (no quantification, is that observed in all ablated inclusions? It does not seem to be true for the second ablated inclusion in the EM picture). The authors attribute RB swelling to ROS action. How would that work? In other words, how would protein or lipid oxidation result in a strong morphology change within 10 min? An osmotic choc is a more likely explanation: the fact that the inclusion retracts when ablated strongly suggests that unlike what textbooks say the osmolarity in the inclusion lumen is different from that in the cytoplasm. Secondly, there is no data showing that the bacteria (EBs) suffer from this. (The fact that RB will not survive inclusion rupture and will not be able to initiate an infection is indisputable). Demonstrating it is certainly not worth the effort (collecting EBs from ruptured inclusions...), but the text needs to be modified (the whole argumentation l248-259 makes a loop on shaky data, and the end of the abstract is far too speculative).

Minor remarks:

3. The authors insist on the point that their method is unique (l.33,112) but do not explain how it differs from other membrane ablation reports.
4. The abstract says that "membrane rupture" and "cell death" are processes that work in concert. The data show that cell death follows membrane rupture, but there is no evidence of two simultaneous pathways.
5. What does "reconcile the pathogen's established capacity to promote host cell survival and induce cell death » mean? These two aspects of Chlamydia developmental cycle have long been

recognized and do not need reconciliation.

6. Awkward use of "egress" l.45 and 46, with an explanation on the second occurrence.
7. Please use *C. trachomatis* or *C. psittaci* (l. 91 and throughout)
8. l.165: I don't think it is surprising that 24 hour old inclusions trigger cell death: they often already contain more than a hundred bacteria, with all their associated PAMPs. The reverse would have been surprising.
9. Fig. 1f, and in many other instances: there is no indication on how many times the experiment was performed, and what percentage of cell death was observed for each ablated object. Were the ablated endosomes always of similar size than the inclusions. Also, the figure is not easy to interpret (took me a while to realize there were two different cells in the field).
10. l. 211: another example of a very expected "finding". In fact, the data show swollen RBs, but do not address the question as to whether infectious particles indeed lose infectivity when exposed to the host cytoplasm, for obvious technical reasons (see above major point 3).
11. l. 262: why is the membrane rupture qualified as "highly coordinated"?
12. The speculation on CADD in the discussion is useless
13. The Methods are not sufficiently detailed. For instance, there is no indication on the concentration used /incubation times for CellROX Green. There is no explanation on how the inclusions were counted, and on which criteria were they classified as "secondary inclusions" rather than "primary inclusions" (there are only examples).
14. Fig1b: the legend mention GFPCTL2, but the images does not show green bacteria
15. Prefer "mCherry" to "Cas9" in the figure legend. Also, I don't understand why there are still so many inclusions at late time points in FigAa DMSO. If those represent "secondary infections", why are they not in mCherry positive cells, since the whole cell culture is supposed to express mCherry.

Reviewer #3 (Remarks to the Author):

In this paper, Kerr et al argue that inclusion rupture triggers a cell death pathway that does not involve apoptosis. This is a major statement since several groups have reported that infected cells displayed apoptotic features, which include TUNEL and Annexin V positive staining, increase in caspase-8 and caspase-3 enzymatic activity, and dependency on pro-apoptotic BAK.

Major comments

1) Fig 3a: The authors use a fluorescence probe that is activated by cas-3/-7. They show activation of this probe in only one cell in which the nucleus was ablated (presumably resulting in apoptosis) but not in cells in which inclusions had been ruptured.

The authors should: 1) show more than one cell with an ablated nucleus; 2) show that zVAD-fmk blocks the activation of the probe; 3) use TUNEL and Annexin V staining to demonstrate that nucleus ablation triggers positive staining whereas inclusion rupture does not.

2) Fig 3b-d: The authors show that knockout of BAX and BAK or treatment with zVAD-fmk (a broad caspase inhibitor) do not affect inclusion rupture-induced survival loss.

To definitively rule out activation of apoptosis the authors should: 1) use a standard caspase enzymatic activity assay (i.e., Asp-Glu-Val-Asp-aminomethylcoumarin (DEVD-AMC)) to demonstrate that cas-3/-7 are not activated following inclusion rupture; 2) check cleavage of cas-3/-7 by Western blot; 3) check BAX/BAK activation by their cleavage, mitochondrial outer membrane insertion, homo-oligomerization); 3) cleavage/activation of the initiator caspase-8 and caspase-9, and the caspase-8 substrate BID (generation of truncated tBID).

Minor comments

p. 5, line 94 – Ref 29 (Ying et al 2006) should appear instead of Ref 10.

p. 5, line 96 – “...which cleaves tBID...” should be “...which cleaves BID to generate truncated tBID...”

p. 14, line 309 – MOMP stands for “Mitochondrial Outer Membrane Permeabilization” and not for “Mitochondrial Outer Membrane Pore”

Point by Point Rebuttal

Reviewer #1 (Remarks to the Author):

Previous studies have shown that the chlamydial inclusion can be extruded from infected cells, and also that cell death of infected cells is affected by the infection. The authors propose the novel view that rupture of the chlamydial inclusion could affect exit of the bacteria from infected cells and also affect host cell death.

The inclusion ablation technique is interesting, but some points would benefit from clarification. The authors write that after inclusion ablation there is "evidence of RB swelling in the ablated inclusions suggestive of a destructive impact of inclusion rupture upon the pathogen", but RBs swell inside intact inclusions when their development is impaired (after treatment of infected cells with IFN γ or adenosine, for example). It's not obvious that the pathogen is being destroyed.

>> Both reviewers 1 and 2 raised similar concerns about our interpretation of the CLEM data following laser ablation. We will document our response to these points collectively here. Firstly, it should be noted that the RB swelling mentioned by reviewer 1 following nutrient deprivation, adenosine- or IFN γ -treatment is a gradual process that takes hours to days to establish as opposed to the very rapid response (minutes) we describe following inclusion disruption. Therefore, the mechanisms inducing these changes are likely to be distinct. We do agree with reviewer 2 that these changes are likely indicative of impaired viability but, to be clear, whilst reviewer 2 stated "*there is no data showing that the bacteria (EBs) suffer from this (swelling)*", we do not suggest that EBs swell in the manuscript. We believe that the robust nature of the EB likely provides some order of protection following inclusion ablation to ensure secondary infection but are unable to establish this. We attempted to examine the impact of ablation upon chlamydial viability and infectivity by monitoring the formation of secondary infections from single cells following inclusion ablation. This assay involved FACS-mediated seeding of individual infected HeLa into uninfected HeLa cell monolayers. These single infected cells were then ablated or allowed to progress through native egress and the subsequent formation of secondary infections (plaques) of neighbouring cells was monitored (see Data Figure *for reviewer consideration only* 1). This assay confirmed that when inclusions were ablated 24 h p.i. the number of viable infectious progeny able to infect neighbouring cells (secondary infections) was much reduced when compared with those allowed to egress naturally. This observation is consistent with early ablation producing less viable bacteria than when natural egress occurs. However, we can't exclude that this reduction simply reflects fewer bacteria present within the inclusion at this earlier stage of its infection cycle and are therefore not willing to include this data. Because CLEM, by nature, is not applicable to population-scale quantification (reviewer 2 point) we agree that more cautious interpretation is required. Accordingly, we have modified the relevant text in that we now focus simply on documenting our observations and have removed some of the more interpretative statements.

Specifically:

Line 223: within this paragraph we have removed:

- "suggestive of a destructive impact of inclusion rupture upon the pathogen"
- "this bactericidal impact may reflect"
- "bactericidal concentrations"

Also, relevant to the CLEM data, we thank the reviewer 2 for pointing towards the possibility of osmotic shock also contributing to RB swelling following inclusion disruption. Grieshaber et al 2002 provides compelling evidence to suggest that the swollen spherical shape of the chlamydial inclusion is maintained by osmotic pressure which would be supportive for osmotic shock playing a role during our ablation experiments. Accordingly we have updated the text to incorporate this possibility.

Specifically:

Line 239 we have added: “Grieshaber et al. (2002) proposed that the swollen shape of the chlamydial inclusion is maintained by osmotic pressure (Grieshaber, Swanson, & Hackstadt, 2002) raising the possibility that inclusion rupture leads to osmotic shock.”

Data Figure *for reviewer consideration only* 1. Single cells infected with GFP-CTL2 were sorted 18 hr p.i. onto a monolayer of uninfected HeLa cells. a) natural egress showing that one infected cell releases a plaque of inclusions 24 hours later. b) examples of the level of secondary infection observed 24 hours after the indicated manipulations.

In addition, the authors need to provide more information regarding the ablation technique in this manuscript, not just refer to their previous manuscript, since it is central to this study. It is not clear why the ablation target is a volume ($0.6 \mu\text{m}^3$), nor why the technique is considered uniformly successful at disrupting the inclusion while never damaging the plasma membrane.

>> We have extended the described methodology as requested. The Z-dimension of our multiphoton imaging system is predicted to be $0.6 \mu\text{m}$ as calculated using a Nyquist rate and PSF calculator (<https://svi.nl/NyquistCalculator>), although the point spread function could be as large as $1-1.5 \mu\text{m}$. Accordingly we selected a region of interest of $0.6 \mu\text{m}^2$ in the X- and Y-axis to complement the predicted $0.6 \mu\text{m}$ in the Z. Under similar conditions Watanabe et al confirmed the plasma membrane integrity when targeting intracellular organelles with femtosecond lasers. To ensure the impact of the ablation was limited to the selected region of interest (ROI), as described in Figure 1B, we monitored the loss of soluble cytoplasmic GFP following inclusion ablation as a means of assessing the integrity of the plasma membrane during laser ablation. In all cases, when the inclusion membrane was targeted we did not see escape of soluble GFP from the cytoplasm until cell death had occurred.

Specifically:

Line 148 we have added: "Optical dissection methods provide the means to locally microirradiate regions of cells at submicron resolutions¹⁶. Unlike long-pulse ultraviolet (UV) and visible lasers, femtosecond lasers that operate in the near infrared region of the spectrum produce efficient two-photon ionization with no out-of-focus absorption¹⁷. Due to nonlinear effects around the focal volume, there is little transfer of heat or mechanical energy to surrounding structures meaning that subcellular organelles may be targeted for photodisruption without influencing underlying or overlying structures. Watanabe et al (2004) demonstrated the efficacy of this approach to selectively ablate individual mitochondria without influencing cellular viability¹⁸."

Line 586 we have updated the text: "The Z-dimension of our multiphoton imaging system is predicted to be $0.6 \mu\text{m}$ as calculated using a Nyquist rate and PSF calculator (<https://svi.nl/NyquistCalculator>), although the point spread function could be as large as $1-1.5 \mu\text{m}$ meaning that the region ablated is estimated to be $\sim 0.6-1.9 \mu\text{m}^3$."

It's not clear why the CellRox would indicate "dramatic accumulation" of ROS in the "inclusion" after ablation. Shouldn't it be similar to the cytosol, but no longer excluded? It is implied that CellRox enters the inclusions, and ROS are excluded.

>> This point was raised by both reviewer 1 & 2. We thank both reviewers for allowing us to clarify this important point, and will document our response collectively at this point. CellROX green is a *membrane permeable* dye and as such would not be excluded from the chlamydial inclusion. The membrane permeability of the CellROX probe is supported by the mitochondrial staining evident within the cytoplasm of the presented cells (arrow heads now added to highlight this for clarity) irrespective of ablation highlighting the capacity for the dye to both cross the plasma membrane and freely enter intracellular organelles. The cell-permeant dye is weakly fluorescent while in a reduced state and exhibits bright green photostable fluorescence upon oxidation by reactive oxygen species (ROS) and subsequent binding to DNA. The elevated CellROX green fluorescence evident within inclusions following ablation must therefore reflect both oxidation and DNA-binding (presumably chlamydial DNA) within the lumen of the ablated inclusions. The most plausible explanation is that

negatively-charged ROSs are excluded by the negatively-charged phospholipid bilayer of the chlamydial inclusion and ablation disrupts this barrier function. We apologise for any confusion caused and have amended the text accordingly to clarify.

Specifically:

Line 251: minor modifications throughout this paragraph

The authors refer to "the so-called Mitochondrial Outer Membrane Pore (MOMP)", but this abbreviation should be avoided if possible to avoid confusion. The chlamydiae also have a "MOMP" (major outer membrane protein) which is well known in the field.

>> We appreciate the confusion caused here and have altered the text accordingly by removing the abbreviation.

The cellular morphology in figure 1a doesn't seem typical of a HeLa cells at 48 hpi; basically the whole cell is occupied by an inclusion in the figure. It is surprising that ablation of a 16 hpi inclusion leads uniformly to cell death within 60 min, while ablation of uninfected cells causes no cell death. It might be informative to do the experiment at an earlier time point, if they can see the inclusions to shoot at.

>> Whilst we observe a range of cellular morphologies at 48 h p.i., the cell presented in 1A (to illustrate a point already established by Hybiske & Stephens (2007)) is not atypical. We have included a time-lapse movie of a population of GFP-CTL2 infected cells from 35 h p.i. as an additional supplementary movie to illustrate this point and reassure Reviewer 1

Specifically:

Line 142: *NEW supplementary movie 2 added.*

In agreement with reviewer 1, we too were surprised that ablation as early as 16 h p.i. was sufficient to lead to cell lysis (albeit with some delay). Whilst attempts were made to ablate inclusions prior to 16 h p.i., the reduced inclusion volume made confirmation of successful inclusion membrane disruption unreliable because the consequent inclusion collapse was less apparent. As such we have limited our observations to this time-point.

The statement that bacterial protein synthesis is not required for the cell lytic process is not fully supported. The authors add chloramphenicol at 24 hpi and observe no difference in host cell lysis at 48 hrs (compared to no chloramphenicol); however, they also observed no difference when they performed ablation at 24 hpi compared to 48 hpi. From the results, it's not certain whether bacterial proteins play a role in subsequent host cell lysis. The authors should tone down this conclusion or substantiate it.

>> We understand reviewer 1's concern and as such we have modified these conclusions as requested.

Specifically:

Line 190 we have updated the text: "Subsequent *de novo* bacterial protein synthesis was not required to trigger cell death as treatment of cells with 100µg/ml chloramphenicol from 24h p.i. did not impact upon the ablation-triggered cell death response (Fig. 1D).

Reviewer #2 (Remarks to the Author):

In this paper, Kerr et al use a laser ablation strategy to control the rupture of the C. trachomatis inclusion membrane, and record ensuing events by live microscopy.

The new findings of this study are:

1. inclusion rupture is inhibited by calpain inhibitors
2. cell-death that follows inclusion rupture does not require integrity of established apoptotic or necroptotic signaling cascades.

My opinion is that these data are too preliminary (point 1) or their interest is too limited (point 2) for publication in Nat. Comm.

Point 1: the calpain data are very preliminary. They only rely on the use of inhibitors, and there is no clue on what cellular process might be implicated. Also, do the inhibitors used target the cysteine-protease activity of calpain, and how specific are they? Comparison of active concentration of the drugs, in combination or not with the cysteine inhibitor used by Hybiske et al should determine whether calpain activity is the only cysteine activity required for inclusion rupture, but even that will not answer the important question: what lies behind this calpain sensitivity?

>> We thank the reviewer for the opportunity to clarify the basis for our conclusions. As the reviewer pointed out, calpains are cysteine proteases and we find that using the best available inhibitor to target calpain cysteine protease activity (calpeptin) in isolation (references included in manuscript) is sufficient to inhibit inclusion rupture. Our previous understanding from the work of Hybiske and Stephens (as cited by the reviewer) was derived using a pan-cysteine protease inhibitor, which non-specifically targets cysteine proteases and thus provided less mechanistic insight than we have reported here, namely that calpains are the crucial enzymes. We feel that *“what lies beyond this calpain sensitivity”* is well beyond this article which already resolves a number of long-standing issues in the field. Indeed, the differences of opinion between the three reviewers regarding what is known and what is not about cell death in egress highlight precisely how important our study is to the field.

Specifically:

Line: 393/492: To address the queries about the specificity of these widely accepted calpain inhibitors, we have added this detail to the relevant results and discussion.

Point 2: While there has been indeed a bit of a debate on whether infected cells undergo apoptosis or not at the end of the infectious cycle most researchers have now agreed for quite some time that apoptotic cascades are not involved (see Curr Immunol Rev. 2007;3(1):31-40 for instance). Finding that sudden release in the cytoplasm of hundreds of bacteria, with all the associated PAMPs, triggers rapid cell death is really not surprising, and likely involves several redundant “hyper-stress” pathways. The authors show that some canonical pro-death signaling pathways are not needed and it will likely be impossible to find a single one that is absolutely required.

>> Firstly, whilst reviewer 2 feels that *“...most researchers have now agreed for quite some time that apoptotic cascades are not involved”* we need only point to reviewer 3 who states *“...cell death pathway that does not involve apoptosis. This is a **major statement** ...”* demonstrating a clear difference of opinion with reviewer 2 in this regard. We believe that this justifies the need to conclusively establish the identity of the death pathway involved.

Secondly, we thank reviewer 2 for providing this additional reference/review, we have incorporated it into our manuscript. In response we would like to establish that whilst this

excellent review article suggests that apoptotic cascades are not involved in *Chlamydia* induced cell death, it does not provide definitive empirical data to support this claim and indeed large portions of the referenced material have proven to be questioned since its publication. Specifically, we point out that a large focus of this particular review is upon how *Chlamydia* induces apoptosis “resistance” through the loss of BH3 proteins, a finding that has subsequently been refuted as artefact by more recently published experimental data (Chen, Johnson, Lee, Sutterlin, & Tan, 2012; Hacker, Heuer, & Ojcius, 2014). Furthermore, this review highlights work which suggests that either BAX inhibitor-1 application or Bcl-2 overexpression can partly revert the death ratio in chlamydia-infected cells, proposing that there may be a mitochondrion-dependent but caspase-independent atypical apoptosis pathway involved. We, however, find that HeLa cells genome-edited to be BAK-null and/or BAX-null die with the same kinetics as the parental line following disruption of the chlamydial inclusion or native egress. Consequently, whilst reviews and expert opinion are an excellent resource, we strongly feel that publication of empirical data that conclusively supports a model, as ours does, is essential.

Thirdly, we are pleased that reviewer 2 fully appreciates a major finding of our manuscript in the induction of several redundant “hyper-stress” pathways leading to cell death. Indeed, we have met strong opposition to this suggestion from those that prefer to suggest that each and every stage of the infectious process is orchestrated by the pathogen directly through the action of specific chlamydial effectors. Indeed, the CADD paragraph, now removed as reviewer 2 felt it was “useless”, alluded to this point. We believe that we have provided compelling data that supports the model that inclusion rupture, and subsequent release of damage and danger signals as referred to by reviewer 2, is indeed the mechanism by which cell death is triggered.

The paper presents a number of shortcomings (see below). In addition, it is not focused on one message, but is more an accumulation of separate questions the authors could address with the laser technology, and fails at delivering a message. As a consequence, it lacks structure. In line with this, the discussion seems randomly organized, does not start with the important message(s), and is hard to follow. For instance the authors chose to make the calpain data the strong point of the various data they collected (see title and running title) but calpain is only introduced (very vaguely, with no reference) line 446 of the discussion, and only the last paragraph of the result section is related.

>> This manuscript is as much about the application of the technology utilised (a first in the field) as the questions it has answered so we feel that structuring the manuscript as we have is appropriate. How a article is written is a matter subject to personal taste. Even though it is clear that all three reviewers fully appreciated the major findings, we have taken the reviewer’s point on board and have sought to clarify our message in the discussion by removing some commentary that does not directly address the primary findings. We thank the reviewer for pointing out that we could more prominently discuss the role of calpains, and have taken the opportunity in revision to do so.

Specifically:

Line 492 added text: “The calpain family constitutes 15 members in humans and can be divided into two subfamilies μ -calpains (activated by subcellular micromolar concentrations of Ca^{2+}) and m-calpains (activated by subcellular millimolar concentrations of Ca^{2+})⁵³. Following autocatalytic hydrolytic activation, calpains translocate to intracellular membranes where they cleave a diverse suite of substrates including cytoskeletal and adhesion proteins, various membrane proteins, kinases, phosphatases, ion transporters and

phospholipases⁵⁴. Notably, calpains have long been associated with various cell death pathways. Calpain activity is required for the release of the Apoptosis-Inducing Factor (AIF) from the mitochondrial intermembrane space where it translocates to the nucleus to mediate caspase-independent, large-scale DNA fragmentation⁴⁶. Furthermore, the aforementioned PARP-1 regulated necroptotic pathway, is dependent upon calpain activity as genomic deletion of Capn4, the regulatory subunit for μ - and m-calpain, completely prevented cell death following treatment with alkylating DNA-damage agents⁴⁶ and TNF α -induced necrosis has also been reported to be dependent on calpain activation⁵⁵. Intriguingly, sustained calpain activation is associated with lysosomal rupture leading to neuronal necrosis postischemia, possibly suggestive of shared mechanisms with the events preceding and following chlamydial inclusion rupture⁵⁶. Further investigation should identify the relevant substrates cleaved by the calpains during chlamydial inclusion rupture and whether they represent a viable target for therapeutic intervention during chlamydial infection.”

Other major remarks

1. Fig. 2c&d show that the CellROX Green probe only reacts with the inclusion lumen content after inclusion ablation. The authors interpret this as a proof that the inclusion membrane protects the bacteria from ROS. A control that the probe reached the inclusion lumen (but did not react) before inclusion ablation is needed (same for the use of Mitotracker Red).

>> please refer to collective response to this point above (reviewer 1, comment 3).

2. The authors claim that inclusion rupture exposes bacteria to ROS, and that this damages the bacteria. The data do not support these conclusions. They only show RB swelling (no quantification, is that observed in all ablated inclusions? It does not seem to be true for the second ablated inclusion in the EM picture). The authors attribute RB swelling to ROS action. How would that work? In other words, how would protein or lipid oxidation result in a strong morphology change within 10 min? An osmotic choc is a more likely explanation: the fact that the inclusion retracts when ablated strongly suggests that unlike what textbooks say the osmolarity in the inclusion lumen is different from that in the cytoplasm. Secondly, there is no data showing that the bacteria (EBs) suffer from this. (The fact that RB will not survive inclusion rupture and will not be able to initiate an infection is indisputable). Demonstrating it is certainly not worth the effort (collecting EBs from ruptured inclusions...), but the text needs to be modified (the whole argumentation l248-259 makes a loop on shaky data, and the end of the abstract is far too speculative).

>> please refer to collective response to this point above (reviewer 1, comment 1).

Minor remarks:

3. The authors insist on the point that their method is unique (l.33,112) but do not explain how it differs from other membrane ablation reports.

>> Our method is *unique* in its application to examining the impact of laser disruption of the limiting membrane of a pathogen containing vacuole and represents a first in the field of chlamydial biology. We thank the reviewer for drawing our attention to possible ambiguity because the methodology has been used in other scenarios, and accordingly have removed this description of our technique from the manuscript.

Specifically:

Original Line 33: Deleted “unique”

Original Line 456: Deleted “unique”

Original line 112: We maintain however, that the approach remains novel in its application.

4. The abstract says that “membrane rupture” and “cell death” are processes that work in concert. The data show that cell death follows membrane rupture, but there is no evidence of two simultaneous pathways.

>> We were referring to the meaning that the two processes *work* together towards the same objective (that is the liberation of the pathogen), not simultaneously as suggested by the reviewer. To avoid any ambiguity, we have altered the wording of the abstract.

Specifically:

Line 39: Changed “work in concert” to “work sequentially”

5. What does “reconcile the pathogen’s established capacity to promote host cell survival and induce cell death » mean? These two aspects of Chlamydia developmental cycle have long been recognized and do not need reconciliation.

>> Whilst they have been *recognised*, we do not feel that they have been *reconciled*. As articulated in detail in our introduction, numerous articles suggest that chlamydia-infection induces apoptotic cell death when it is also established that infected cells are resistant to both extrinsic and intrinsic apoptotic stimuli. We have provided mechanistic insight that conclusively demonstrates that the cell death pathway is not dependent on the previously reported apoptotic mechanisms. This reconciles these incompatible observations by demonstrating that chlamydia infected cell death occurs via non-apoptotic pathways rather than the cell death pathway that the infection perturbs. We appreciate that reviewer 2 feels that this is established, but given reviewer 3’s response, this is evidently not the case.

6. Awkward use of “egress” l.45 and 46, with an explanation on the second occurrence.

>> Changed as requested.

Original Line 46: Deleted “or egress”.

7. Please use *C. trachomatis* or *C. psittaci* (l. 91 and throughout)

>> This has been changed throughout as requested.

8. l.165: I don’t think it is surprising that 24 hour old inclusions trigger cell death: they often already contain more than a hundred bacteria, with all their associated PAMPs. The reverse would have been surprising.

>> We note that this opinion is at odds with reviewer 1 (and our own) who found cell death following early ablation “surprising”. However, upon reflection we have removed “surprising” from the text.

Original Line 185: Deleted “Surprisingly, a”

9. Fig. 1f, and in many other instances: there is no indication on how many times the experiment was performed, and what percentage of cell death was observed for each ablated object. Were the ablated endosomes always of similar size than the inclusions. Also, the figure is not easy to interpret (took me a while to realize there were two different cells in the field).

>> Endosomes of similar dimensions to the ablated inclusions were always selected for concurrent ablation for comparison. We deliberately selected a representative example (from 10 experiments conducted) presenting the minimal interpretable number required (ie. a single infected cell and a single uninfected cell). We had thought it obvious there were two cells by the indicated two nuclei (highlighted by arrows) and the fact that one cell died and the other did not. To make this clearer, we have numbered the 2 cells respectively and these details have been added and clarification provided.

10. l. 211: another example of a very expected “finding”. In fact, the data show swollen RBs, but do not address the question as to whether infectious particles indeed lose infectivity when exposed to the host cytoplasm, for obvious technical reasons (see above major point 3).

>> We have incorporated this response above in response to reviewer 1, point 1 who also made similar comments.

11. l. 262: why is the membrane rupture qualified as “highly coordinated”?

>> We do not describe membrane rupture as “highly coordinated” at any stage. The consequent cell death, post-rupture, is highly coordinated in that it was consistently observed to occur in a robustly reproducible time-frame following inclusion ablation.

12. The speculation on CADD in the discussion is useless

>> The motivation for its inclusion is described earlier but we have removed the speculative discussion about this chlamydial effector protein.

Original Line 383: Deleted text “Notably, *C. trachomatis* L2 does encode the *Chlamydia* protein associating with death domains (CADD) (Stenner-Liewen et al., 2002). Whilst this protein is reported to bind and activate elements of the host’s apoptotic pathway *in vitro*, its function has never been examined *in vivo*. It would be interesting to monitor the rate of egress and inclusion rupture-triggered cell death in CADD-null strains should they prove viable.”

13. The Methods are not sufficiently detailed. For instance, there is no indication on the concentration used /incubation times for CellROX Green. There is no explanation on how the inclusions were counted, and on which criteria were they classified as “secondary inclusions” rather than “primary inclusions” (there are only examples).

>> We have corrected the minor concerns identified regarding the depth of detail provided in the methods. We have sought to improve the level of detail included keeping in mind the broad readership.

14. Fig1b: the legend mention GFPCTL2, but the images does not show green bacteria

>> The reference to GFPCTL2 has been removed for clarity.

15. Prefer “mCherry” to “Cas9” in the figure legend. Also, I don’t understand why there are still so many inclusions at late time points in FigAa DMSO. If those represent “secondary infections”, why are they not in mCherry positive cells, since the whole cell culture is supposed to express mCherry.

>> Firstly, as described in the methods these cells express the fusion protein mCherry-Cas9 so the suggested change in the figure legend is inappropriate. Secondly, we would ask the reviewer to look more carefully at the data presented in figure 4A. The time-lapse video-microscopy clearly demonstrates that there are <10 intact inclusions remaining in the DMSO-treated samples (reduced from hundreds), most of which are in surviving mCherry-Cas9 treated cells. We can only assume that the reviewer has confused ruptured cellular debris as intact chlamydial inclusions.

Reviewer #3 (Remarks to the Author):

In this paper, Kerr et al argue that inclusion rupture triggers a cell death pathway that does not involve apoptosis. This is a **major statement since** several groups have reported that infected cells displayed apoptotic features, which include TUNEL and Annexin V positive staining, increase in caspase-8 and caspase-3 enzymatic activity, and dependency on pro-apoptotic BAK.

Major comments

1) Fig 3a: The authors use a fluorescence probe that is activated by cas-3/-7. They show activation of this probe in only one cell in which the nucleus was ablated (presumably resulting in apoptosis) but not in cells in which inclusions had been ruptured.

The authors should: 1) show more than one cell with an ablated nucleus; 2) show that zVAD-fmk blocks the activation of the probe; 3) use TUNEL and Annexin V staining to demonstrate that nucleus ablation triggers positive staining whereas inclusion rupture does not.

>> We thank reviewer 3 for these useful control experiments. We have conducted the requested experiments and have included them as supplementary figures and movies as, whilst we appreciate their importance, we do not feel they contribute to the main thrust of the manuscript and as such do not warrant inclusion as primary figures.

Specifically, 1) Supplementary movie 13 demonstrates a field of view in which the nuclei of 5 cells have been ablated. As is apparent all of them activate the CellEvents probe (line 302). 2) Application of zVad-fmk was sufficient to inhibit activation of the probe (see supplementary movie 14) however it is difficult to draw any conclusions based upon this observation as the cells no longer die (consistent with them dying via an apoptotic mechanism). We have limited our interpretation of these particular observations as a consequence (line 307).

As presented in Supplementary Figure 4 (line 325), laser-ablation of nuclei resulted in a localised cauterisation wound within the DNA of cell’s that was prominently TUNEL-labelled. This labelling extended in a more diffuse manner throughout the entire nuclei of these cells consistent with them having initiated apoptosis and DNA fragmentation. Whilst there was no evidence of TUNEL-labelling within the nuclei of cells in which inclusions had been

ablated (or neighbouring cells not targeted), a striking accumulation of TUNEL-labelling was observed within *Chlamydia* in which inclusions had been disrupted. This supports the premise that inclusion disruption exposes the bacteria within to the likely hostile cytoplasm of host cells.

2) Fig 3b-d: The authors show that knockout of BAX and BAK or treatment with zVAD-fmk (a broad caspase inhibitor) do not affect inclusion rupture-induced survival loss.

To definitively rule out activation of apoptosis the authors should: 1) use a standard caspase enzymatic activity assay (i.e., Asp-Glu-Val-Asp-aminomethylcoumarin (DEVD-AMC)) to demonstrate that cas-3/-7 are not activated following inclusion rupture; 2) check cleavage of cas-3/-7 by Western blot; 3) check BAX/BAK activation by their cleavage, mitochondrial outer membrane insertion, homo-oligomerization); 3) cleavage/activation of the initiator caspase-8 and caspase-9, and the caspase-8 substrate BID (generation of truncated tBID).

>> Thank you for these suggestions. 1) The CellEvents probe used in Figure 3A uses the identical DEVD cleavage site as the DEVD-AMC assay suggested but unlike DEVD-AMC, it is compatible with standard microscopes. As such we do not feel that conducting this experiment would provide any additional insight. 2) Unfortunately the laser ablation approach is not compatible with population-scale analyses like western immunoblot. Instead we have conducted correlative live cell and immunofluorescent microscopy in single cells to examine cleavage of caspase 3 using a specific antibody. Consistent with the CellEvents probe, disruption of the chlamydial inclusion did not induce cleavage of caspase 3 (see Supplementary Figure 3) (line 315). 3) We have conducted ablation experiments in infected cells expressing GFP-BAX. There was no evidence of mitochondrial membrane insertion following inclusion ablation. See Supplementary Movie 18 (line 360). Similar to the strategy described for cleaved caspase 3, we have conducted correlative live cell and immunofluorescence microscopy to examine cleavage of caspase 8 using a specific antibody. Again, there is little evidence of cleaved caspase 8 in cells in which the chlamydial inclusion has been disrupted (see Supplementary Figure 3)(line 322). Overall these additional experiments are consistent with our earlier observations that activation of apoptosis is not occurring.

Minor comments

p. 5, line 94 – Ref 29 (Ying et al 2006) should appear instead of Ref 10.

>> Original Line 94: Thank you. This reference has been corrected.

p. 5, line 96 – “...which cleaves tBID...” should be “...which cleaves BID to generate truncated tBID...”

>> Original Line 96: Thank you for picking up this oversight.

p. 14, line 309 – MOMP stands for “Mitochondrial Outer Membrane Permeabilization” and not for “Mitochondrial Outer Membrane Pore”

>> We have made this change and additionally have removed the abbreviation at the request of reviewer 1 to save on confusion with the chlamydial major outer membrane protein “MOMP”.

- Chen, A. L., Johnson, K. A., Lee, J. K., Sutterlin, C., & Tan, M. (2012). CPAF: a Chlamydial protease in search of an authentic substrate. *PLoS Pathog*, 8(8), e1002842. doi:10.1371/journal.ppat.1002842
- Grieshaber, S., Swanson, J. A., & Hackstadt, T. (2002). Determination of the physical environment within the *Chlamydia trachomatis* inclusion using ion-selective ratiometric probes. *Cell Microbiol*, 4(5), 273-283.
- Hacker, G., Heuer, D., & Ojcius, D. M. (2014). Is the hoopla over CPAF justified? *Pathog Dis*, 72(1), 1-2. doi:10.1111/2049-632x.12211
- Konig, K. (2000). Robert Feulgen Prize Lecture. Laser tweezers and multiphoton microscopes in life sciences. *Histochem Cell Biol*, 114(2), 79-92.
- Stenner-Liewen, F., Liewen, H., Zapata, J. M., Pawlowski, K., Godzik, A., & Reed, J. C. (2002). CADD, a *Chlamydia* protein that interacts with death receptors. *J Biol Chem*, 277(12), 9633-9636. doi:10.1074/jbc.C100693200
- Watanabe, W., Arakawa, N., Matsunaga, S., Higashi, T., Fukui, K., Isobe, K., & Itoh, K. (2004). Femtosecond laser disruption of subcellular organelles in a living cell. *Opt Express*, 12(18), 4203-4213.

REVIEWERS' COMMENTS:

Reviewer #1 (Remarks to the Author):

The authors have addressed my previous concerns.

Reviewer #2 (Remarks to the Author):

My initial main comments on this report was that the data were too preliminary (inclusion rupture is inhibited by calpain inhibitors) or their interest was too limited (cell-death that follows inclusion rupture does not require integrity of established apoptotic or necroptotic signaling cascades) for publication in Nat. Comm.. The paper was not substantially modified during revision, thus my opinion remains unchanged, and this is a matter of editorial decision. However, there are still two experimental shortcomings that were not addressed satisfactorily during revision, no matter the journal.

1 - One important claim of the paper is to demonstrate that ROS are excluded from the inclusion until the inclusion membrane is ruptured. This claim is based on the observation that a "membrane permeant dye" CellROX Green, which binds to DNA and fluoresces upon oxidation, does not stain the bacteria when the inclusion membrane is intact.

I see two big problems with this experiment:

-the first problem, that was also pointed out by reviewer 1, is that there is no demonstration that the probe reaches the inclusion lumen (and the bacteria) until the inclusion membrane is ruptured. The fact that the probe is sold as "cell permeant" is not sufficient; the inclusion membrane could have different properties than the plasma membrane, which is used to define "cell permeability". In fact, I believe that the experiment with the Mitotracker Red CMXRos demonstrates the limit of these probes. Mitotracker Red is also sold as a "membrane permeant" dye, that, when oxidized, becomes conjugated to thiol groups. In eukaryotic cells it stains mitochondria because it is where most O₂ is produced. Like bacteria-derived mitochondria, Chlamydia produce a lot of O₂ in a rather small volume, and should also be strongly stained with Mitotracker, if the probe could reach the bacteria. The fact that Mitotracker Red stains the bacteria only once the inclusion membrane is ruptured demonstrates that the probe Mitotracker does not diffuse through the inclusion membrane, at least not at concentrations that allow oxidation and thiol-conjugation in the bacteria. The same might well be true for the other probe, CellROX Green.

- If we assume that CellROX Green is able to reach the inclusion lumen, we face a second problem: the ROS indirectly observed accumulating in the inclusion upon inclusion membrane ablation could very well be produced because of the trauma of the ablation technology itself. In other words, there is no evidence that the ROS detected in the inclusion 10 min post ablation were already present in the host cytoplasm before ablation.

Thus, the statement "we now provide direct evidence that the chlamydial inclusion provides a protective barrier to the hostile cytoplasm, potentially protecting the delicate RBs within from cytosolic ROS" » I269 is incorrect (also l. 116 etc) , as is the conclusion of the discussion (l. 523).

2 - The whole paragraph « Premature inclusion rupture is detrimental to chlamydial development » does not report anything more than RB swelling upon membrane rupture (the authors agree that we cannot tell what causes this swelling. They persist in invoking the implication of ROS, but without providing any link between ROS and swelling. And most of all, this whole paragraph does not show what it claims (« detrimental to chlamydial development ») for obvious reasons that I had already anticipated in my earlier review, and that the authors acknowledge in their letter: it is technically impossible to see if EB suffer from membrane rupture. As for the RBs, disruption of the inclusion membrane, which results in cell lysis within 30 min, I don't see how this could not be « detrimental to chlamydial development », since they will never be able to make it to the infectious EB stage. Thus the whole paragraph needs to be deleted (the authors may want to keep their

observation of RB swelling somewhere, but this is anecdotic).

Reviewer #3 (Remarks to the Author):

The authors have adequately addressed all my comments.

RESPONSE TO REVIEWERS' COMMENTS:

Reviewer #1 (Remarks to the Author):

The authors have addressed my previous concerns.

Reviewer #2 (Remarks to the Author):

My initial main comments on this report was that the data were too preliminary (inclusion rupture is inhibited by calpain inhibitors) or their interest was too limited (cell-death that follows inclusion rupture does not require integrity of established apoptotic or necroptotic signaling cascades) for publication in Nat. Comm.. The paper was not substantially modified during revision, thus my opinion remains unchanged, and this is a matter of editorial decision. However, there are still two experimental shortcomings that were not addressed satisfactorily during revision, no matter the journal.

1 - One important claim of the paper is to demonstrate that ROS are excluded from the inclusion until the inclusion membrane is ruptured. This claim is based on the observation that a “membrane permeant dye” CellROX Green, which binds to DNA and fluoresces upon oxidation, does not stain the bacteria when the inclusion membrane is intact.

I see two big problems with this experiment:

- the first problem, that was also pointed out by reviewer 1, is that there is no demonstration that the probe reaches the inclusion lumen (and the bacteria) until

the inclusion membrane is ruptured. The fact that the probe is sold as “cell permeant” is not sufficient; the inclusion membrane could have different properties than the plasma membrane, which is used to define “cell permeability”. In fact, I believe that the experiment with the Mitotracker Red CMXRos demonstrates the limit of these probes. Mitotracker Red is also sold as a “membrane permeant” dye, that, when oxidized, becomes conjugated to thiol groups. In eukaryotic cells it stains mitochondria because it is where most O₂ is produced. Like bacteria-derived mitochondria, Chlamydia produce a lot of O₂ in a rather small volume, and should also be strongly stained with Mitotracker, if the probe could reach the bacteria. The fact that Mitotracker Red stains the bacteria only once the inclusion membrane is ruptured demonstrates that the probe Mitotracker does not diffuse through the inclusion membrane, at least not at concentrations that allow oxidation and thiol-conjugation in the bacteria. The same might well be true for the other probe, CellROX Green.

- If we assume that CellROX Green is able to reach the inclusion lumen, we face a second problem: the ROS indirectly observed accumulating in the inclusion upon inclusion membrane ablation could very well be produced because of the trauma of the ablation technology itself. In other words, there is no evidence that the ROS detected in the inclusion 10 min post ablation were already present in the host cytoplasm before ablation. Thus, the statement “we now provide direct evidence that the chlamydial inclusion provides a protective barrier to the hostile cytoplasm, potentially protecting the delicate RBs within from cytosolic ROSs » l269 is incorrect (also l. 116 etc) , as is the conclusion of the discussion (l. 523).

2 - The whole paragraph « Premature inclusion rupture is detrimental to chlamydial development » does not report anything more than RB swelling upon membrane rupture (the authors agree that we cannot tell what causes this swelling. They persist in invoking the implication of ROS, but without providing any link between ROS and swelling. And most of all, this whole paragraph does not show what it claims (« detrimental to chlamydial development ») for obvious reasons that I had already anticipated in my earlier review, and that the authors acknowledge in their letter: it is technically impossible to see if EB suffer from membrane rupture. As for the RBs, disruption of the inclusion membrane, which results in cell lysis within 30 min, I don't see how this could not be « detrimental to chlamydial development », since they will never be able to make it to the infectious EB stage. Thus the whole paragraph needs to be deleted (the authors may want to keep their observation of RB (swelling somewhere, but this is anecdotic).

>> We thank the reviewer for there feedback. We do not believe that there is any evidence to suggest that the inclusion membrane is particularly impermeant to the probes utilised as it is derived from the host cell membranes through the hijacking of membrane trafficking pathways. Indeed, as articulated in our manuscript, Grieshaber¹ et al. reported that that the lumen of the chlamydial inclusion shares most biophysical properties in common with the cytoplasm and there is a free exchange of cytoplasmic ions. We do not feel that it is necessary to remove this data entirely as we feel it is informative and worth sharing with the research community however we appreciate reviewer 2's concerns and as such we have markedly toned down the conclusions

drawn from the CellROX and MitoTracker experiments instead using them to simply focus on the barrier function of the inclusion rather than any destructive influence the molecules they report upon may have upon the pathogen. We have also modified the text to include reviewer 2's suggestion that the probes are excluded from the inclusion.

1. Grieshaber, S., Swanson, J.A. & Hackstadt, T. Determination of the physical environment within the *Chlamydia trachomatis* inclusion using ion-selective ratiometric probes. *Cell Microbiol* **4**, 273-283 (2002)

Reviewer #3 (Remarks to the Author):

The authors have adequately addressed all my comments.